# Post-replicative initial expression of *PAX6* during neuroectoderm differentiation

Song Hu [ID][1,2,3,7], Rongao Kou [ID][4,7], Zhuojie Su[4], Guanchen Li[1,2,3], Shutao Qi [ID][2,3], Yanxiao Zhang [ID][3,4], Haifeng Wang[5], Ling-Ling Chen[6] & Hongtao Yu [ID][2,3,4 ✉]

## Abstract

**The development of multicellular organisms requires precise coordination between cell division and differentiation. Cell division generates the necessary number of cells, while differentiation creates distinct cell identities, forming tissues and organs. The transcription factors SOX2 and PAX6 specify neuroepithelial cells, the earliest neural progenitor cells (NPCs) during brain development. How lineage specification is coordinated with the cell cycle is not fully understood. Here, we show that *PAX6* expression occurs during a narrow time window after neural induction of human embryonic stem cells (ESCs). Flow cytometry analyses and time-lapse imaging show that *PAX6* expression starts during the G2 phase. We identify a novel 500-bp *PAX6* promoter that drives its G2-specific expression. *PAX6* expression is independent of known regulators of cell cycle-dependent transcription, suggesting the existence of a novel mechanism. S-phase block by hydroxyurea prevents *PAX6* expression and differentiation into NPC. Thus, NPC fate specification is coupled to cell cycle progression and occurs after the completion of DNA replication. This post-replicative lineage commitment ensures the creation of two daughter cells of identical cell fate following cell division.**

**Keywords** Cell Cycle; Cell Fate Decision; Transcription; Neurogenesis; Neural Progenitor Cells
**Subject Categories** Cell Cycle; Chromatin, Transcription & Genomics; Development

## Introduction

The development of multicellular organisms requires precisely timed and spatially coordinated processes that balance cell division and differentiation. Cell division generates enough cells necessary for building tissues and organs, while differentiation assigns distinct identities to these cells, enabling functional specialization (Belmonte-Mateos and Pujades, 2021). This balance is dynamically regulated by extracellular signals, such as WNT, BMP, SHH, and FGF families of proteins, which form morphogen gradients to instruct cells based on their spatial positioning (Kicheva and Briscoe, 2023). These signals activate intracellular pathways that regulate transcription factor networks, which in turn govern cell identity (Spitz and Furlong, 2012).

For differentiation into terminally differentiated, functional cell types that no longer divide, such as neurons, the precursor cells often undergo cell cycle exit and activate the expression of cell-fate-determining genes to commit to a particular cell type (Buttitta and Edgar, 2007; Ruijtenberg and van den Heuvel, 2016). During early embryogenesis, however, pluripotent stems cells (PSCs) differentiate into multipotent precursors of specialized lineages that can themselves undergo active cell divisions (Liu et al, 2019; Rossant and Tam, 2022). In this context, it is unclear whether cell fate decisions are coupled to cell cycle progression, and if so, during which cell cycle phase the lineage commitment is made.

The mammalian nervous system develops from neuroepithelial cells, the earliest neural progenitor cells (NPCs) lining the neural tube (Kelley and Pasca, 2022). These cells exhibit apical-basal polarity and undergo symmetric divisions to expand the progenitor pool before giving rise to radial glia that can both self-renew and undergo asymmetric divisions to produce neurons (Coquand et al, 2024; Kelley and Pasca, 2022). The transcription factors (TFs) SOX2 and PAX6 regulate neuroectodermal specification (Mercurio et al, 2022; Ochi et al, 2022; Zhou et al, 2016). Mutations in SOX2 and PAX6 cause severe eye and brain malformations (Hever et al, 2006; Mercurio et al, 2022; Ochi et al, 2022).

The cell cycle is tightly regulated by phase-specific transcriptional programs (Fischer et al, 2022). Cell cycle-dependent transcription is mediated by conserved regulatory elements in gene promoters, which recruit TFs in a temporally controlled manner. The DREAM complex (DP, RB-like, E2F, and MuvB) is a key transcriptional repressor of mitotic genes. During cell cycle progression, the DREAM complex is disrupted, leading to the formation of the related MMB-FOXM1 complex that contains B-MYB and FOXM1 (Fischer et al, 2022). The MMB-FOXM1

[1]College of Life Sciences, Zhejiang University, Hangzhou, Zhejiang, China. [2]New Cornerstone Science Laboratory, School of Life Sciences, Westlake University, Hangzhou, Zhejiang, China. [3]Westlake Laboratory of Life Sciences and Biomedicine, Hangzhou, Zhejiang, China. [4]School of Life Sciences, Westlake University, Hangzhou, Zhejiang, China. [5]School of Life Sciences, Center for Synthetic and Systems Biology, Tsinghua-Peking Joint Center for Life Sciences, Tsinghua University, Beijing, China. [6]New Cornerstone Science Foundation, Key Laboratory of RNA Innovation, Science and Engineering, CAS Center for Excellence in Molecular Cell Science, Shanghai Institute of Biochemistry and Cell Biology, University of Chinese Academy of Sciences, Chinese Academy of Sciences, Shanghai, China. [7]These authors contributed equally: Song Hu, Rongao Kou. ✉E-mail: yuhongtao@westlake.edu.cn

complex binds to promoters of genes encoding mitotic regulators, such as cyclin B, PLK1, and Aurora kinases, through interactions with cell cycle homology region (CHR) elements or CCAAT-box motifs (Fischer et al, 2016; Müller et al, 2012; Schmit et al, 2009). At the G2–M transition, CDK1 and PLK1 phosphorylate and activate the MMB-FOXM1 complex, promoting the transcription of mitotic genes (Branigan et al, 2021; Fischer et al, 2022).

Emerging evidence suggests that the cell cycle is not merely a passive timer but an active regulator of differentiation. In mammalian systems, PSCs lengthen their cell cycle, particularly the G1 phase, during differentiation (Calder et al, 2013; Calegari and Huttner, 2003; Lange et al, 2009; Salomoni and Calegari, 2010). G1 cyclins, such as cyclin D1–3, alter the propensity of PSCs to differentiate into different lineages through regulating the TGFβ pathway. It has been proposed that cell cycle-dependent mechanisms restrict the activation of developmental genes in PSCs to the G1 phase (Dalton, 2015).

Previous work has established an in vitro differentiation protocol to differentiate PSCs into NPCs by inhibiting TGFβ and BMP pathways (Chambers et al, 2009). In this study, using this dual SMAD inhibition (dSMADi) differentiation protocol, we differentiated human embryonic stem cells (ESCs) into NPCs and monitored the expression of the NPC marker PAX6 during the process using flow cytometry and live-cell imaging of cells expressing FUCCI (Fluorescent Ubiquitination-based Cell Cycle Indicator) (Pauklin and Vallier, 2013; Sakaue-Sawano et al, 2008). *PAX6* activation is confined to a narrow temporal window (48 to 72 h after differentiation initiation). Its initial expression coincides with the G2 phase of the cell cycle. Intriguingly, a 500-base pair (bp) region of the *PAX6* promoter drives its G2-specific expression in a mechanism that does not appear to involve the DREAM complex. S-phase arrest by the DNA replication inhibitor hydroxyurea abolishes PAX6 expression and blocks NPC specification. These findings reveal a novel mechanism where NPC commitment is contingent on cell cycle progression, with lineage determination occurring post-replication to ensure the generation of two NPCs upon division.

# Results

## Neuroectoderm cell fate transition occurs in a short time window during in vitro differentiation

To investigate the cell fate transition of neuroectoderm in vitro, we differentiated H9 ESCs into NPCs using the established dSMADi neural induction protocol (Chambers et al, 2009), which required 18 days to differentiate ESCs into NPCs. To assess the robustness of this neural induction system, we collected cells at various timepoints during differentiation and subjected them to Western blotting (Appendix Fig. S1A). As SOX2 was expressed highly in both ESCs and NPCs (Sivakumar et al, 2022; Zhou et al, 2016), the established NPC marker PAX6 was used to track NPC identity. The PAX6 protein was absent in ESCs and accumulated during neural induction. We then characterized the transcriptional profiles of differentiated NPCs using bulk RNA sequencing (RNA-seq). Pathway analysis revealed that, compared to ESCs, upregulated differentially expressed genes (DEGs) in NPCs at day 18 (D18) were predominantly associated with central nervous system

development (Appendix Fig. S1B), confirming successful NPC generation. To evaluate their differentiation potential, we differentiated NPCs into forebrain neurons. Immunofluorescence (IF) demonstrated widespread expression of the neuronal marker TUJ1 in NPC-derived cells post-differentiation (Appendix Fig. S1C). RNA-seq showed that upregulated DEGs in differentiated cells were linked to synaptic formation and neurogenesis (Appendix Fig. S1D), validating functional neuronal differentiation. Collectively, these results confirm the robust transition from ESCs to NPCs in our in vitro system, enabling subsequent mechanistic studies of cell fate transitions.

Next, we sought to pinpoint the initial timing of NPC fate specification. Samples were collected daily during the first 6 days of neural induction and analyzed for PAX6 expression with Western blotting, IF, and Fluorescence-Activated Cell Sorting (FACS). PAX6 protein levels rose sharply between D2 and D3 (Fig. 1A), corroborated by IF and FACS quantification showing PAX6+ cells increasing from ~5% at D2 to ~50% by D3 (Fig. 1B–D). To confirm whether this time window indeed corresponded to the ESC-to-NPC transition, we also examined pluripotency markers (NANOG and OCT4) in addition to PAX6 by IF. NANOG disappeared by D2, while OCT4 persisted at low levels on D2 but was absent by D3 (Fig. 1E). PAX6 expression initiated on D2 and further increased by D3. Thus, D2 and D3 represent the critical phase of NPC fate specification.

To probe gene expression dynamics during this transition, we performed bulk RNA-seq and mass spectrometry to profile transcriptomic and proteomic changes. RNA-seq revealed extensive transcriptional remodeling between D2 and D3 (1037 upregulated and 1834 downregulated DEGs; Fig. EV1A). Proteomic analysis identified fewer differentially expressed proteins (247 upregulated and 302 downregulated proteins; Fig. EV1B). Both upregulated genes and proteins were enriched in neurodevelopmental pathways (Fig. EV1C,D). The transcript-protein concordance was partial (Fig. EV1E,F). Notably, *PAX6* expression exhibited concurrent upregulation patterns in both transcriptomic and proteomic datasets (Fig. 1F), suggesting that its transcriptional activation might contribute to PAX6 expression after D2 of neural induction.

Time-series proteomic clustering revealed four distinct protein expression patterns during ESC-to-NPC differentiation (Fig. EV1G). Clusters 2 and 3, representing upregulated proteins, were enriched in neural development pathways, again consistent with neural induction (Fig. EV1H). The four clusters were also enriched for additional pathways, including ion transport, RNA processing, and cell cycle regulation. These proteomic changes highlight the complexity of cell fate transitions and the involvement of multiple cellular processes, including the cell cycle.

## PAX6 expression initiates in G2 during NPC cell-fate transition

We next analyzed PAX6 expression from D2 to D3 during neural induction. Western blotting revealed a gradual increase in PAX6 protein levels during differentiation from 48 to 72 h (Fig. 2A; Appendix Fig. S2A). Quantitative PCR demonstrated a similar upregulation of *PAX6* mRNA levels (Appendix Fig. S2B). FACS analysis further confirmed a continuous increase in the proportion of PAX6+ cells over this period (Fig. 2B; Appendix Fig. S2C). Interestingly, the earliest PAX6+ cells detected by FACS had 4 N DNA content, indicating that they were in G2/M (Fig. 2B). To

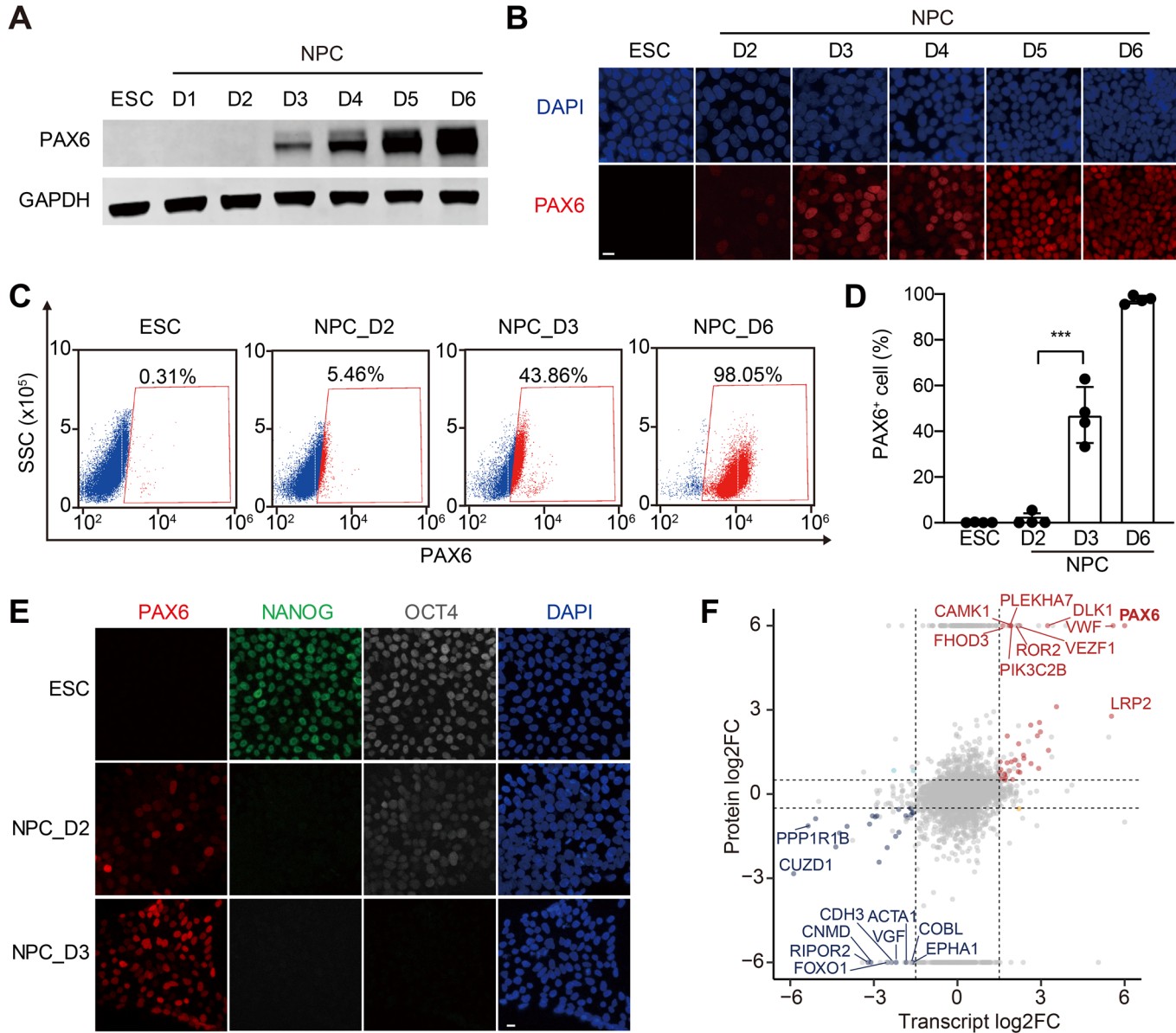

**Figure 1. The ESC–NPC cell fate transition occurs between day 2 (D2) and D3 of neural induction.**

(A) Western blot analysis of PAX6 protein levels in ESCs and NPCs during D1 to D6 of neural induction. GAPDH was used as the loading control. (B) Images of ESCs and NPCs at indicated times of neural induction stained with anti-PAX6 antibody and DAPI. Scale bar, 10 μm. (C) FACS analysis of the percentage of PAX6+ cells in ESCs and NPCs at indicated times of neural induction. (D) Quantification of the percentage of PAX6+ cells in (C). Mean ± SD; $P$ value (***) = 0.0003 (paired $T$ test); $n = 4$ independent experiments. (E) Images of ESCs and NPCs at D2 and D3 of neural induction stained with DAPI and antibodies against NANOG, OCT4, and PAX6. Scale bar, 10 μm. (F) Scatter plot showing the correlation of transcriptomics and proteomics data of differentially expressed genes and proteins between NPC_D3 and NPC_D2. The horizontal and vertical dotted lines indicate significance boundaries (transcriptomics: $P$ value < 0.01 and |log2FC| >1.5; proteomics: $P$ value < 0.05 and |log2FC| >0.5). The $t$ test is applied to the proteomic data. The transcriptomic data are analyzed with the negative binomial generalized linear model (GLM) with a Wald test for significance and the Benjamini-Hochberg procedure to control the false discovery rate. Source data are available online for this figure.

further pinpoint whether PAX6 expression initiated in G2 or mitosis, we employed the FUCCI system to simultaneously monitor PAX6 expression and the cell cycle status from 46 to 72 h using IF (Fig. 2C; Appendix Fig. S2D). PAX6+ cells first emerged at 46 h of differentiation, with their percentages gradually increasing. Strikingly, nearly all PAX6+ cells at early timepoints were GEMININ+ and CDT1− with intact nuclei, indicating that these cells were in G2 (Fig. 2C,D).

To determine whether *PAX6* mRNA also emerged during G2, we performed single-cell RNA sequencing (scRNA-seq) to profile transcriptional dynamics during ESC-to-NPC differentiation. UMAP clustering revealed distinct distributions of ESC, NPC_24h, NPC_72h, and NPC_D6 populations, whereas NPC_46h to NPC_54h samples clustered together and exhibited overlapping profiles (Fig. 2E), consistent with the gradual progression of neural induction during that time window. Analysis of marker genes

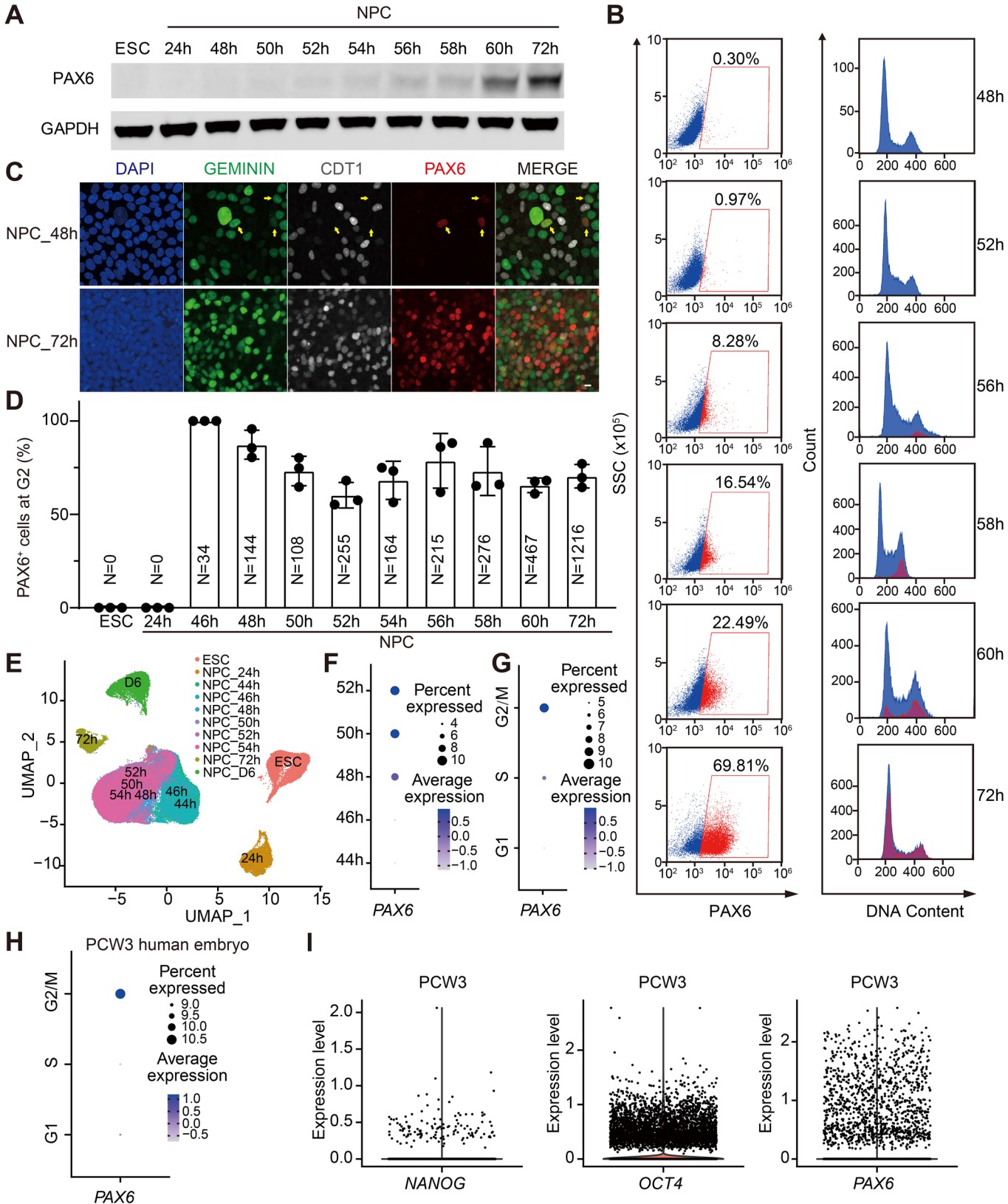

**Figure 2. PAX6 expression is activated at G2 phase during ESC–NPC cell fate transition.**

(A) Western blot analysis of PAX6 protein levels in ESCs and NPCs at different timepoints (24–72 h) of neural induction. GAPDH was used as the loading control. (B) FACS analyses of the percentage of PAX6+ cells (left panels) and the cell cycle status (right panels) in NPCs at different timepoints (48–72 h) during neural induction. (C) Images of FUCCI NPCs at 48 and 72 h of neural induction stained with DAPI and the PAX6 antibody (red). The cell cycle status was determined by the FUCCI reporters (GEMININ, green; CDT1, white). The yellow arrows indicate newly generated PAX6+ cells that are GEMININ-positive and CDT1-negative. Scale bar, 10 μm. (D) Quantification of the percentage of PAX6+ cells at the G2 phase in (C). Data are presented as mean ± SD, $n = 3$ independent experiments. N indicates the total number of PAX6+ cells counted in each group from the three independent experiments. (E) UMAP plot of single-cell RNA sequencing (scRNA-seq) data of ESCs and NPCs at different times of neural induction. (F) Dot plot showing PAX6 expression at different timepoints (44–52 h) of neural induction based on the scRNA-seq data in (E). (G) Dot plot showing the cell cycle distribution of all PAX6+ cells during 24–52 h of neural induction based on integrated scRNA-seq data. (H) Dot plot showing the cell cycle distribution of all PAX6+ cells in publicly available scRNA-seq datasets of post-conceptional week (PCW) 3 human embryonic tissues; $n = 1$ embryo. (I) Violin plot showing the gene expression of NANOG, OCT4, and PAX6 in datasets described in (H). Source data are available online for this figure.

showed rapid downregulation of ESC markers NANOG and OCT4 and progressive PAX6 upregulation upon neural induction (Appendix Fig. S3A). SOX2, expressed in both ESCs and NPCs, was markedly elevated in NPCs, consistent with prior studies (Zhang et al, 2019; Zhou et al, 2016). The percentage of PAX6-expressing cells gradually increased between 48 and 52 h of differentiation (Fig. 2F). Cell cycle analysis revealed that PAX6 expression was predominantly enriched in G2/M (Fig. 2G; Appendix Fig. S3B). Thus, PAX6 mRNA accumulation preferably occurred during G2.

Other neural development genes also exhibited cell cycle-specific expression patterns during neural induction. For example, MAFK and FOXA2 were mainly expressed in G1 (Appendix Fig. S3C), whereas TEAD2 and ZEB1 were expressed in G2 (Appendix Fig. S3D). These findings suggest that G2-enhanced transcription is not unique to PAX6, but represents a broader regulatory mechanism for multiple genes during cell fate transitions.

In human embryos, PAX6 expression can be detected as early as post-conceptional week (PCW) 3 (Zhang et al, 2010). To explore the timing of PAX6 expression in vivo, we analyzed publicly available scRNA-seq datasets from human embryos. Our analysis revealed that cells with PAX6 expression were predominantly enriched in G2 at PCW3, a developmental stage characterized by low NANOG expression and persistent OCT4 expression (Fig. 2H,I) (Zeng et al, 2023). When we analyzed PCW4 human embryos, we found that both NANOG and OCT4 were no longer expressed, whereas PAX6 expression was increased (Appendix Fig. S3E,F). Interestingly, PAX6 expression remained enriched in G2 at this stage. Therefore, consistent with our in vitro findings, G2-enriched PAX6 expression is also observed during neuroectoderm specification in human embryos.

### PAX6 expression activates downstream transcriptional events

Previous studies have identified PAX6 target genes in mature NPCs (Bhinge et al, 2014; Coutinho et al, 2011; Sun et al, 2015; Thakurela et al, 2016; Xie et al, 2013). To characterize PAX6 target genes in early neural induction, we generated two PAX6 KO ESC lines and performed bulk RNA-seq at NPC_D3 (Appendix Fig. S4A–C). DEG analysis showed that genes significantly downregulated in PAX6_KO_NPC_D3 were primarily enriched in pathways related to forebrain development, including PAX6, LHX2, HES5, ASCL2, and LHX9 (Appendix Fig. S4C). This supports a critical role of PAX6 in activating forebrain-specific transcriptional programs even at the early stages of neural induction.

We next performed Cleavage Under Targets and Tagmentation (CUT&Tag) assays to identify genes bound by PAX6 in NPC_D3 samples. Many PAX6-occupied genes were involved in axon guidance, such as LMO3, PAX6, TCF7L2, and CDH2 (Appendix Fig. S4D). Among them, 45 genes were downregulated in PAX6 KO NPC_D3 cells, suggesting that they might be direct PAX6 targets during early neural induction (Appendix Fig. S4E). Integration with single-cell RNA-seq data revealed that, among these putative PAX6 target genes, PAX6, MECOM, and CDH6 exhibited transcript enrichment in the G2 phase of NPC_48h cells (Appendix Fig. S4F,G). This finding suggested that these genes might be directly activated in the G2 phase by PAX6 when it first emerged. Collectively, these results support a functional role of PAX6 expression in G2 during neural differentiation.

### A novel 500-bp promoter drives G2 PAX6 expression

PAX6 is known to use three promoters (P0, P1, and Pα) for expression, with P1 implicated in early neural development (Aota et al, 2003; Grocott et al, 2007). For a silenced gene to become active, chromatin remodeling must increase the accessibility of its promoter for transcription factor binding (Li et al, 2007). We performed Assay for Transposase-Accessible Chromatin using sequencing (ATAC-seq) to map chromatin accessibility changes during neural induction (Fig. 3A) (Minnoye et al, 2021). The chromatin accessibility of promoters of OCT4 and NANOG greatly decreased by D1 of neural induction, consistent with their role in maintaining pluripotency (Fig. EV2A). Genome-wide analysis identified numerous chromatin regions transitioning from closed to open states during neural induction, with associated genes enriched for neural development pathways (Fig. EV2B). Notably, the accessibility of the PAX6 P1 promoter region markedly increased between D2 and D3 (Fig. 3A), consistent with the expression timing of PAX6. This finding suggested that P1 might mediate cell cycle-dependent PAX6 transcription.

We next engineered a lentiviral reporter system in ESCs, with the expression of the miRFP680 fluorescence protein driven by the P1 promoter or P1-containing upstream regions (1 kb, 2 kb, or 5 kb; termed P1-1 kb, P1-2 kb, P1-5 kb). Surprisingly, all P1-containing constructs showed expression in both ESCs and NPCs (Fig. EV2C,D). Thus, P1-based promoters could not recapitulate NPC-specific expression of the endogenous PAX6. We found that a 192-bp region of the 410-bp P1 promoter overlapped with the first exon of PAX6. This region is known to be bound by OCT4 and NANOG in ESCs (Barakat et al, 2018; Kashyap et al, 2009), providing a potential explanation of the untimely reporter expression in ESCs.

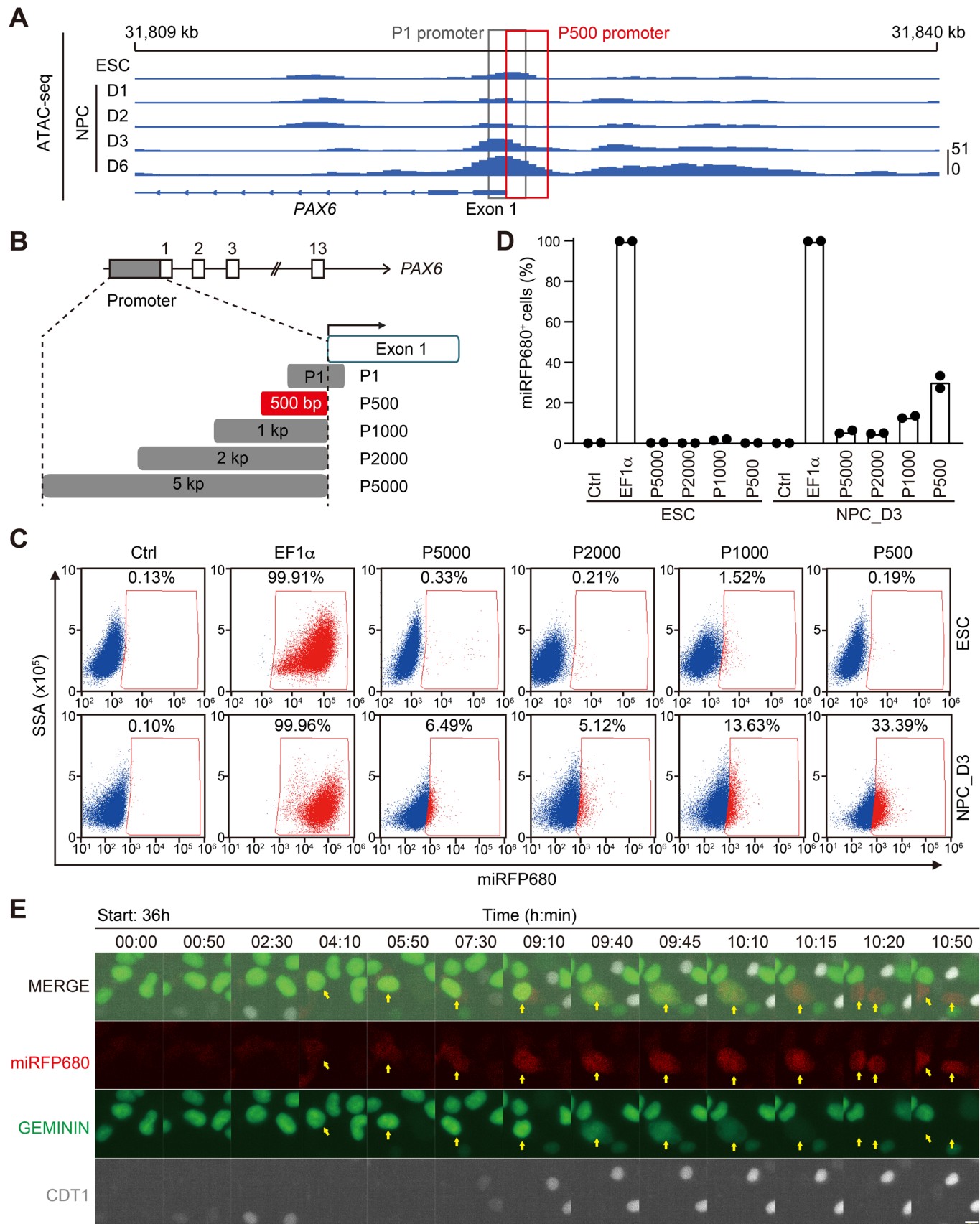

**A** — P1 promoter / P500 promoter. ATAC-seq tracks for ESC and NPC (D1, D2, D3, D6) across the *PAX6* locus (31,809 kb – 31,840 kb). Exon 1 indicated.

**B** — *PAX6* gene structure with exons 1, 2, 3, 13. Promoter constructs: P1, P500 (500 bp), P1000 (1 kp), P2000 (2 kp), P5000 (5 kp) relative to Exon 1.

**C** — Flow cytometry plots of SSA (×10⁵) vs miRFP680 for Ctrl, EF1α, P5000, P2000, P1000, P500 in ESC and NPC_D3.

ESC: Ctrl 0.13%, EF1α 99.91%, P5000 0.33%, P2000 0.21%, P1000 1.52%, P500 0.19%

NPC_D3: Ctrl 0.10%, EF1α 99.96%, P5000 6.49%, P2000 5.12%, P1000 13.63%, P500 33.39%

**D** — miRFP680⁺ cells (%) for Ctrl, EF1α, P5000, P2000, P1000, P500 in ESC and NPC_D3.

**E** — Start: 36h. Time (h:min). MERGE, miRFP680, GEMININ, CDT1 channels at 00:00, 00:50, 02:30, 04:10, 05:50, 07:30, 09:10, 09:40, 09:45, 10:10, 10:15, 10:20, 10:50.

◄ **Figure 3. A novel *PAX6* promoter activates initial *PAX6* expression during the ESC–NPC transition.**

(A) ATAC-seq tracks showing chromatin accessibilities at the *PAX6* locus of ESCs and NPCs from D1 to D6 of neural induction. The P1 promoter and the newly identified 500-bp promoter (P500) of *PAX6* are highlighted with grey and red rectangles, respectively. (B) Schematic diagram of the P1 and candidate *PAX6* promoters tested in reporter assays. (C) FACS plots quantifying the percentage of cells with miRFP680 expression driven by indicated *PAX6* promoters in ESCs and NPCs at D3 of neural induction (NPC_D3). EF1α was used as the positive control and uninfected cells were used as the negative control (Ctrl). (D) Quantification of the percentage of miRFP680$^+$ cells in (C). $n = 2$ independent experiments. (E) Live-cell imaging of FUCCI NPCs expressing the miRFP680 reporter (red) driven by the P500 promoter during 36–46 h of neural induction. Cell cycle status is indicated by the FUCCI reporter system (GEMININ, green; CDT1, white). Yellow arrows mark miRFP680$^+$ cells. Scale bar, 10 μm. Source data are available online for this figure.

We then tested upstream regions (500 bp, 1 kb, 2 kb, 5 kb; termed P500, P1000, P2000, P5000) adjacent to the *PAX6* transcriptional start site (TSS) (Fig. 3B). These reporters showed negligible activity in ESCs but varying levels of activation in NPC_D3, with P500 exhibiting the highest activity (Fig. 3C,D). The lower expression of the reporter driven by larger promoters could be due to the presence of repressor elements or different lentivirus titer/transducing efficiency, with the larger plasmids potentially having lower titers. Regardless, these data suggested that the P500 promoter is the minimal promoter for *PAX6* transcription in NPCs. To determine if the P500-driven reporter expression was dependent on the cell cycle, we introduced the P500 reporter into FUCCI-expressing ESCs and simultaneously tracked its expression and cell cycle status during neural induction. Live-cell imaging analysis identified 90 cells exhibiting miRFP680 reporter activation, all of which initiated reporter expression at G2 (Fig. 3E), mirroring endogenous *PAX6*. The P500-driven miRFP680 reporter was not expressed in 293FT cells (Fig. EV2E), suggesting regulation by TFs specific to the neural lineage. Together, these results identify the P500 promoter as the functional promoter driving cell cycle-coupled *PAX6* transcription during NPC specification.

We next tested whether the G2-enhanced *PAX6* transcription was caused by increased chromatin accessibility of the *PAX6* promoter by performing single-cell ATAC-seq (scATAC-seq) on ESCs and differentiating NPCs. Consistent with the bulk ATAC-seq result, the promoter regions of *OCT4* and *NANOG* gradually closed during NPC differentiation (Appendix Fig. S5A), while the *PAX6* P500 promoter was already open in ESCs and its chromatin accessibility markedly increased in D3 and D6 NPCs (Fig. 3A; Appendix Fig. S5B). The chromatin accessibility around the P500 promoter did not, however, dramatically increase between the critical time window of 48–54 h after neural induction. We then performed an integrative analysis of scATAC-seq and scRNA-seq data to assess chromatin accessibility across different cell cycle phases and found no significant difference in the accessibility of the *PAX6* P500 promoter among G1, S, and G2/M phases, regardless of whether the cells were *PAX6*-positive or -negative (Appendix Fig. S5C). Thus, the enhanced *PAX6* transcription in G2 does not appear to be driven by increased chromatin accessibility.

## G2-enhanced transcription of *PAX6* is independent of the MMB-FOXM1 complex

The MuvB core component of the DREAM complex interacts with BMYB and FOXM1 to form the MMB-FOXM1 complex, which coordinates gene expression during S and G2/M (Fischer et al, 2022). We investigated whether the MMB-FOXM1 complex contributed to *PAX6* transcriptional activation in G2. Most G2-expressed genes, such as *CCNB1*, contain cell cycle-dependent

elements (CDEs) or cell cycle gene homology regions (CHRs) in their promoters (Müller et al, 2012). Analysis of the *PAX6* P500 promoter revealed a putative CHR site predicted to bind LIN54 and FOXM1 (Fig. 4A).

To assess the potential role of these factors, we used CRISPR-Cas9 technology to knock out *FOXM1* and *LIN54* in ESCs. Two *FOXM1* homozygous knockout (KO) clones were successfully generated using two different sgRNAs (Fig. EV3A). In contrast, multiple attempts to knockout *LIN54* with three different sgRNAs yielded clones with frameshift mutations in one allele and non-frameshift mutations in the other, suggesting that *LIN54* might be essential for ESC viability and could not be fully deleted (Fig. EV3B). Despite this, all KO clones exhibited reduced cyclin B1 levels (Fig. 4B), and FACS analysis revealed aberrant cell cycle profiles in *FOXM1*- or *LIN54*-deficient cells, with increased G2/M populations (Fig. 4C,D), confirming their roles in G2 regulation and mitotic entry.

We next assessed whether *FOXM1* or *LIN54* inactivation influenced PAX6 expression during NPC fate transition. In both *FOXM1* KO and *LIN54* KO lines, the percentage of PAX6$^+$ cells in NPC_D2 was not substantially altered, with the initial expression of PAX6 still occurring in G2 (Fig. 4E,F). Thus, *PAX6* transcriptional activation in G2 appears to be independent of key components of the MMB-FOXM1 complex.

## A 12-bp GC-rich motif is required for *PAX6* transcription during NPC cell fate transition

We next investigated the molecular mechanism driving *PAX6* expression during G2. We first examined the potential involvement of several reported TFs of *PAX6*, including SEF (TFCP2), SP1, and SIX3 (Liu et al, 2006; Zheng et al, 2001). Bulk RNA-seq data indicated that *TFCP2* and *SP1* were already expressed in ESCs, whereas *SIX3* was absent in ESCs and began to be expressed from D1 of neural induction (Fig. EV3C). The level of the SIX3 protein was greatly elevated on D2 of neural differentiation (Fig. EV3D), prior to *PAX6* activation. Moreover, SIX3 has been reported as a top hit in a previous CRISPR screen of *PAX6* regulators, while TFCP2 and SP1 were not identified in that screen (Wu et al, 2022). Finally, our scRNA-seq analysis revealed that *SIX3* transcription was also enriched in G2 (Fig. EV3E). These findings suggested SIX3 as an attractive candidate of upstream *PAX6* transcriptional regulators during NPC cell fate transition. We then knocked out *SIX3* in ESCs and obtained two KO cell lines (Fig. EV3F). PAX6 protein expression was, however, not altered in these *SIX3* KO lines (Fig. EV3G), indicating that SIX3 was not required for *PAX6* expression.

We thus decided to further identify critical cis-regulatory elements of *PAX6*. We generated a series of 50-bp truncations within the P500 promoter (designated P500_Δ1 to Δ10) and

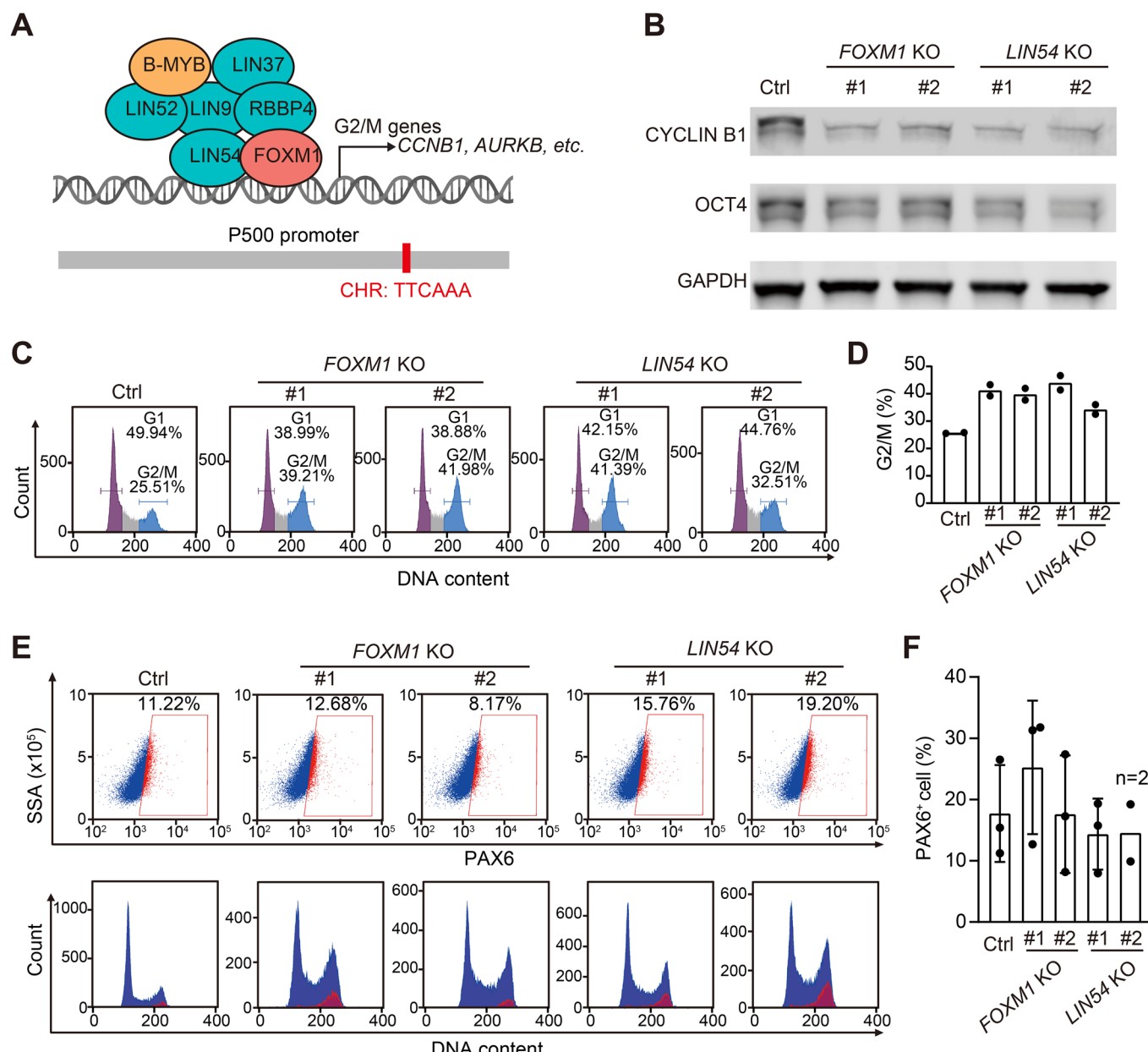

**Figure 4. The DREAM/MMB components FOXM1 and LIN54 are not required for initial *PAX6* expression during ESC-NPC transition.**

(A) Schematic illustration of the MMB-FOXM1 complex that drives G2/M-specific gene expression. The *PAX6* P500 promoter contains a putative CHR (cell cycle genes homology region) motif that may be bound by FOXM1 and LIN54. (B) Western blots showing the protein levels of CYCLIN B1 and OCT4 in wild-type (WT), *FOXM1* KO, and *LIN54* KO ESC clones. GAPDH was used as the loading control. (C) FACS analysis of the cell cycle status of WT, *FOXM1* KO, or *LIN54* KO ESCs. The percentages of cells in G1 and G2/M are indicated. (D) Quantification of G2/M percentages of cells in (C). $n = 2$ independent experiments. (E) FACS analysis of PAX6 expression (upper panels) and the cell cycle status (lower panels) of WT, *FOXM1* KO, and *LIN54* KO NPCs at 56 h of neural induction. (F) Quantification of PAX6+ cells in (E). $n = 3$ independent experiments for WT, *FOXM1* KO clone #1 and #2, and *LIN54* KO clone #1. $n = 2$ independent experiments for *LIN54* KO clone #2. Source data are available online for this figure.

delivered these constructs into ESCs using lentiviruses (Fig. 5A). FACS analysis revealed that none of the reporter constructs were expressed in undifferentiated ESCs. Upon differentiation into NPCs, the intact P500 promoter robustly activated the expression of miRFP680. Fluorescence signals were most dramatically reduced in cells carrying the P500_Δ2 promoter (Fig. 5B,C), implicating the second 50-bp region in transcriptional activation. Consistent with

our findings that the MMB-FOXM1 complex components were not required for *PAX6* expression, deletion of the 8th 50-bp region, which contained the putative CHR motif, did not reduce the expression of PAX6 during NPC formation.

Using the JASPAR database, we predicted TF binding sites within the P500 promoter (Appendix Table S1). Interestingly, 33 of the top 50 predicted TFs targeted a 12-bp GC-rich sequence within

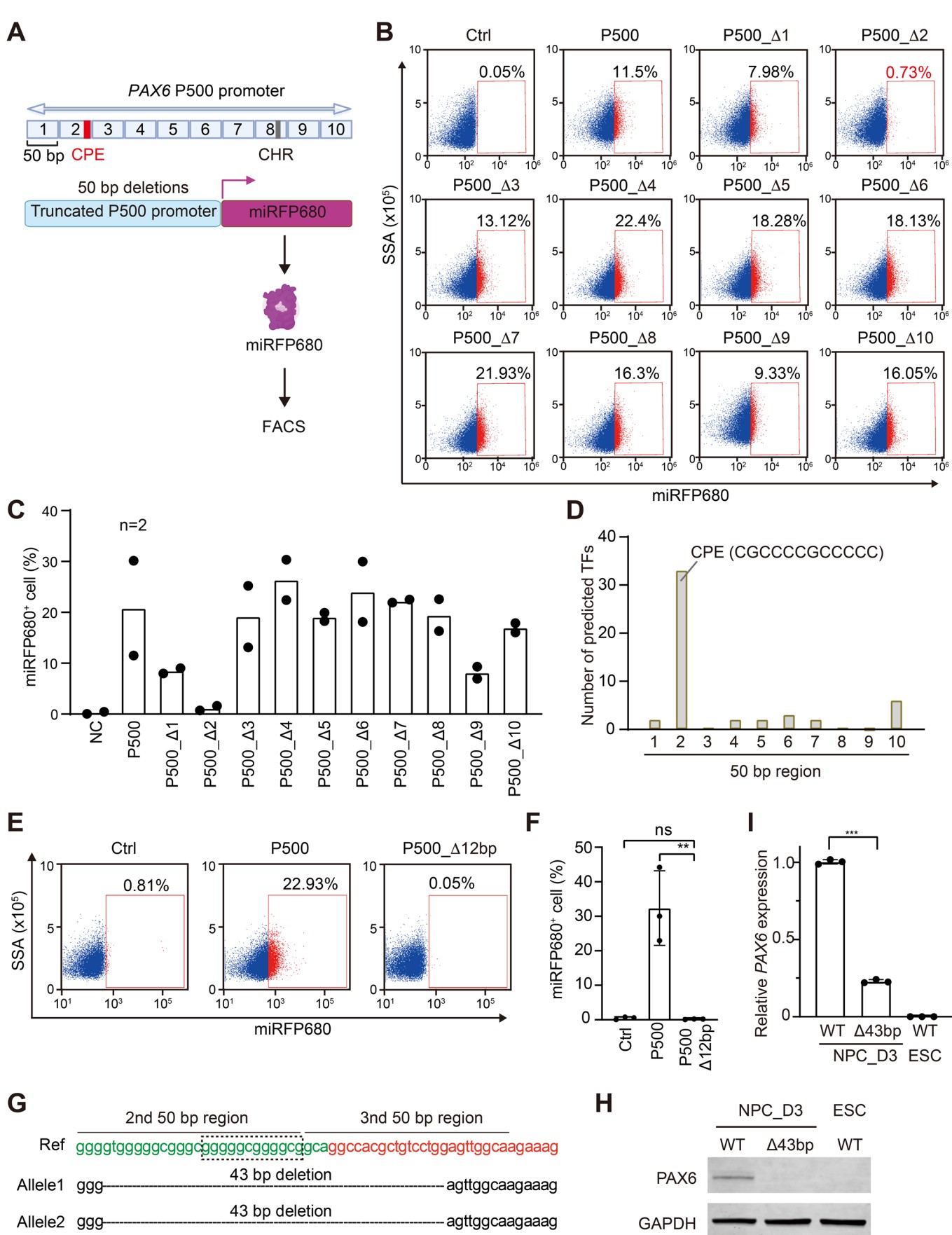

◀ **Figure 5. The core promoter element (CPE) of the P500 promoter is required for initial *PAX6* expression during the ESC–NPC transition.**

(A) Schematic illustration of the miRFP680 reporter assay driven by truncated P500 promoters in NPCs. The P500 promoter was divided into ten 50-bp segments. Each fragment was deleted from the P500 promoter to generate a series of truncated promoters termed P500_Δ1, P500_Δ2, P500_Δ3, and so on. The core promoter element (CPE) identified in this study and the putative CHR motif are indicated. (B) FACS analysis of the percentage of miRFP680$^+$ cells in NPCs at day 3 of neural induction. The miRFP680 reporter was driven by the truncated P500 promoters described in (A). Uninfected cells were used as the negative control (Ctrl). (C) Quantification of miRFP680$^+$ cells in (B). $n = 2$ independent experiments. (D) Distribution of the top 50 predicted TFs in each 50-bp segment of the P500 promoter. The TF-binding consensus sequence in the 2nd segment is shown. (E) FACS analysis of miRFP680 expression in NPCs expressing the miRFP680 reporter driven by the P500 promoter with the 12-bp CPE deleted (P500_Δ12 bp) at day 3 of neural induction. (F) Quantification of the percentage of miRFP680$^+$ cells in (E). $n = 3$ independent experiments; ns: $P > 0.05$, **$P < 0.001$, Student's unpaired two-tailed $t$ test. (G) Sanger sequencing results showing the 43-bp deletion in the endogenous *PAX6* promoter in the Δ43 bp ESC line. DNA sequences of the 2nd and 3rd 50-bp segments of the P500 promoter are in green and red fonts, respectively. The 12-bp CPE is boxed. (H, I) Western blotting (H) and qPCR (I) analyses of the expression of PAX6 protein and transcripts in ESCs and NPCs at day 3 of neural induction (NPC_D3). NPCs were derived from WT ESCs or ESCs with the 43-bp deletion in the endogenous *PAX6* promoter. Mean ± SD; $n = 3$ independent experiments in (I). Source data are available online for this figure.

the second 50-bp region (Fig. 5D), suggesting this motif as a core promoter element regulating *PAX6* transcription. Indeed, the P500_Δ12 bp promoter lacking the 12-bp sequence could not activate the transcription of the reporter in NPC_D3 (Fig. 5E,F).

We next attempted to delete the 12-bp motif from the endogenous *PAX6* promoter using CRISPR/Cas9. Due to technical challenges posed by the high GC content surrounding this motif, we could only obtain an ESC line with a 43-bp deletion spanning this region on both alleles (Δ43 bp ESCs; Fig. 5G). While Δ43 bp ESCs maintained pluripotency marker expression and normal morphology (Fig. EV4A), PAX6 induction was severely impaired upon differentiation to NPC_D3 (Fig. 5H,I). These results identify a novel GC-rich core promoter element (CPE) essential for *PAX6* transcription during NPC fate transition.

We next sought to identify key TFs regulating *PAX6* transcriptional activation during G2 in NPC fate transition. Using the JASPAR database, we selected the top five predicted TFs that bind the CPE for analysis (Appendix Table S1). *EGR1* expression was elevated during ESC–NPC differentiation and preceded that of *PAX6* (Fig. EV4B). Single-cell RNA-seq further showed *EGR1* transcript enrichment at G2/M (Fig. EV4C). The EGR1 protein was present in ESCs and throughout NPC differentiation (up to NPC_D6; Fig. EV4D). We thus tested whether EGR1 regulated *PAX6* transcription. Using CRISPR-Cas9, we generated *EGR1* KO ESC lines (Fig. EV4E). On D3 of neural induction, *EGR1* KO clones showed undetectable EGR1 protein, but the protein levels of PAX6 were not substantially altered (Fig. EV4F,G), indicating that EGR1 was not required for PAX6 expression. Individual deletion of other predicted TFs—including PATZ1, ZNF454, and SP8—also had no effect on cell cycle-dependent *PAX6* expression during NPC fate transition. Thus, PAX6 expression might be regulated by a novel TF or by the redundant actions of the tested TFs.

## Blocking the cell cycle at S phase prevents PAX6 expression

To investigate the biological significance of G2 transcriptional activation in NPC fate transition, we blocked the cell cycle prior to the G2 phase during differentiation using hydroxyurea (HU), a DNA synthesis inhibitor that arrested cells in S phase. We administered 250 μM HU to the culture medium starting on D1 or D2 of differentiation, with treatment durations of 48 or 24 h, respectively, culminating at D3 (Fig. 6A). FACS analysis revealed that DMSO-treated control cells maintained normal cell cycle progression, with 55.2% cells expressing PAX6 by D3 (Fig. 6B,C). In contrast, HU-treated cells exhibited prolonged S-phase arrest

and a complete absence of PAX6 expression (Fig. 6B,C). IF analysis further demonstrated that HU-treated cells failed to express PAX6 by D3 and displayed enlarged nuclei compared to controls (Fig. 6D). These findings indicate that S-phase arrest disrupts normal NPC fate transition, either by preventing progression along the same trajectory or by altering differentiation trajectories.

To distinguish between these two possibilities, we performed transcriptomic profiling across differentiation stages with or without HU treatment. Principal component analysis (PCA) revealed a clear trajectory for untreated cells from D1 to D4, whereas HU-treated cells—regardless of treatment duration—clustered distinctly from D3 controls. The 48-h HU treatment induced greater divergence than 24-h treatment, suggesting that cell cycle arrest fundamentally perturbs differentiation rather than merely delaying it (Fig. 6E).

Transcriptome comparison of D3 HU-treated (24-hour duration) and untreated cells highlighted divergent pathways. Control differentiating NPC_D3 cells upregulated genes enriched in neurodevelopmental pathways as compared to NPC_D2 (Fig. EV5A), whereas HU-treated cells activated mesenchymal lineage programs, including urogenital, renal, and muscular systems (Fig. EV5B). Direct comparison of treated versus untreated D3 cells confirmed upregulation of mesenchymal signature genes (e.g. *TNNT2*, *ACTA1*, *MYOCD*, *ITGA11*, and *FOXC1*) and suppression of neurodevelopmental pathways (e.g. *NR2F1*, *CDH6*, *SP8*, *GLI3*, and *EMX2*) (Fig. 6F–H).

Among the significantly enriched GO terms (adjusted $P < 0.05$) that were upregulated following HU treatment, we did not observe pathways related to cellular stress responses. Heatmap analysis showed that several S-phase-related E2F target genes, including *CCNE1*, *CDC25A*, *CDK2*, *CDC6*, and *POLA2*, were upregulated following HU-induced arrest (Fig. 6H) (Bracken et al, 2004). These results suggest that HU treatment does not interfere with the transcription of canonical S-phase genes. The observed cellular state alterations of HU-arrested cells are likely the consequence of an S-phase block, not non-specific stresses.

We then removed HU from NPC_D3_HU_24h samples and cultured them in the absence of HU for another 24 h (NPC_D4_HU_release). In parallel, NPC_D3_Ctrl samples were allowed to undergo neural induction under standard conditions for an additional 24 h, yielding NPC_D4_Ctrl samples. Bulk RNA-seq analysis of these samples revealed that, compared to NPC_D3_Ctrl, genes upregulated in NPC_D4_Ctrl were predominantly enriched in pathways related to brain development, including *PAX6*, *SP8*, *NR2F1*, *OTX1*, and *EMX2* (Fig. EV5C,D), consistent with the expected neurodevelopmental trajectory. Interestingly, genes

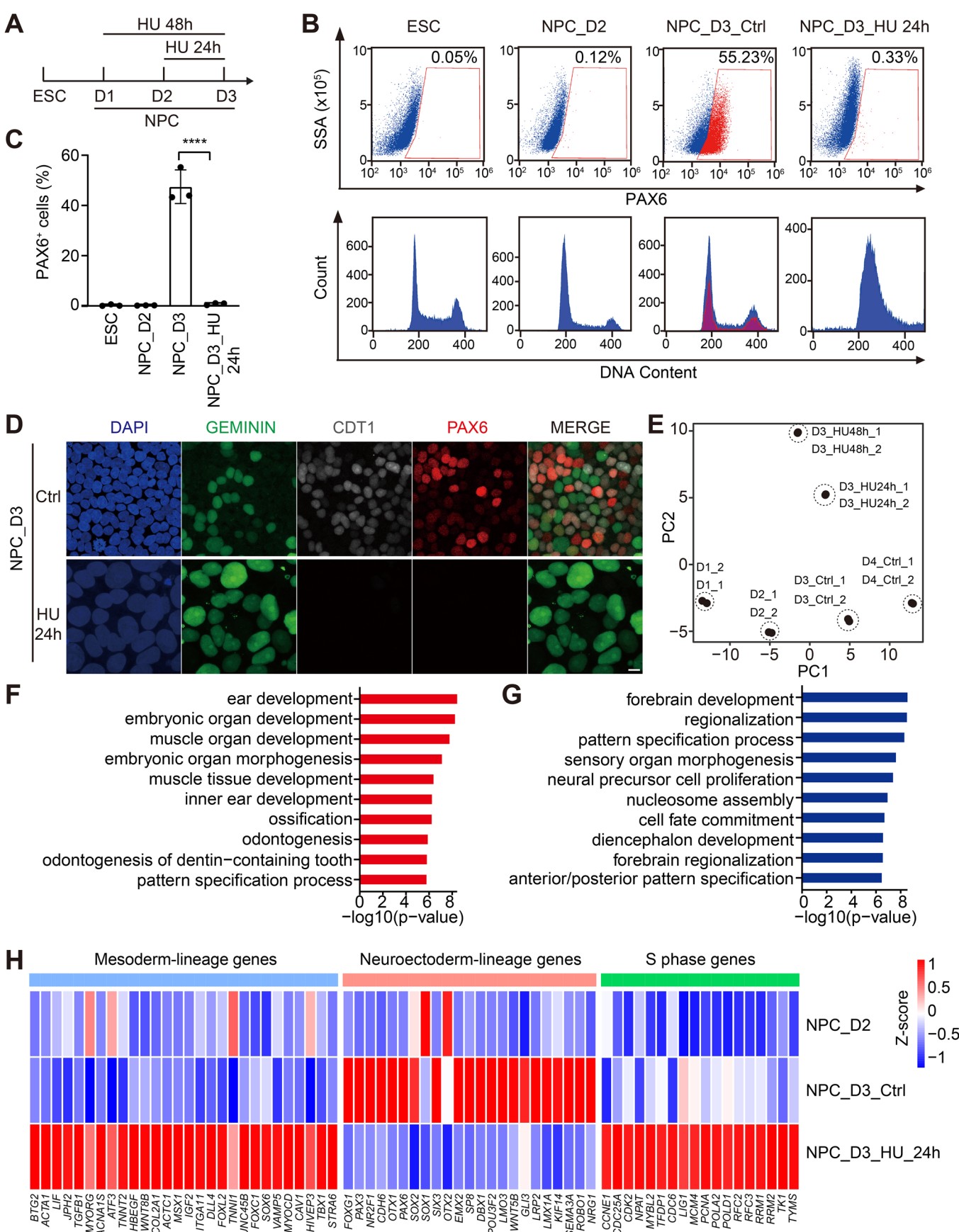

**Figure 6.   S phase arrest blocks *PAX6* expression and ESC–NPC transition.**

(A) Experimental scheme of hydroxyurea (HU) treatment during neural induction. (B) FACS analysis of PAX6 expression (upper panels) and cell cycle status (lower panels) of ESCs and NPCs at day 2 (NPC_D2) or day 3 (NPC_D3) of neural induction, with DMSO (Ctrl) or HU treatment. (C) Quantification of the percentage of PAX6+ cells in (B). Mean ± SD; $n = 3$ independent experiments. (D) Images of FUCCI NPCs at D3 of neural induction with or without HU treatment stained with DAPI and the PAX6 antibody (red). The cell cycle status was determined by FUCCI reporters (GEMININ, green; CDT1, white). Scale bar, 10 μm. (E) Principal component analysis (PCA) of transcriptomic profiles of cells at various differentiation stages, with or without HU treatment. (F, G) Gene Ontology (GO) enrichment analysis of upregulated (F) and downregulated (G) DEGs in HU-treated NPC_D3 cells compared to DMSO-treated NPC_D3 cells. Top ten pathways are shown. (H) Heatmap showing mesoderm-lineage, neuroectoderm-lineage, and S-phase transcribed genes (E2F target genes) in NPC_D2, NPC_D3 with or without HU treatment. The color bar represented Z-scores. Source data are available online for this figure.

upregulated in NPC_D4_HU_release samples compared to NPC_D3_HU_24h samples were also enriched in brain development pathways (e.g. *PAX6*, *SP8*, *FOXG1*, *CDH6*, and *EMX2*) and in pathways associated with mitotic nuclear division (e.g. *MKI67*, *CENPF*, *PLK1*, *CCNB1*, and *AURKA*) (Fig. EV5D,E). These findings suggest that, following the release from HU-induced S-phase arrest, NPCs could resume cell cycle progression and restore the neural differentiation trajectory. We note, however, that the expression levels of neurodevelopmental genes in NPC_D4_HU_release samples did not reach those observed in NPC_D4_Ctrl, indicating that the deviation from the neurogenic trajectory caused by HU treatment could not be fully reversed following the removal of HU. Taken together, these results suggest a requirement for G2 phase progression in NPC fate transition. S-phase arrest not only halts neural differentiation but also redirects cells toward mesenchymal lineages, underscoring the critical role of cell cycle dynamics in dictating progenitor cell fate.

## Discussion

In this study, using an ESC–NPC in vitro differentiation system and PAX6 as the NPC fate marker, we demonstrate that the transition from ESCs to NPCs occurs during the G2 phase. Our findings further suggest that the G2 phase is required for successful NPC fate determination, highlighting the critical role of the cell cycle in lineage commitment. Mechanistically, we identify a novel 500-bp *PAX6* promoter (P500) responsible for cell cycle-dependent transcriptional activation. Interestingly, the previously reported *PAX6* P1 promoter in neural tissues drives untimely *PAX6* expression in ESCs, suggesting the existence of negative regulators in ESCs that act through the P500 promoter. Systematic dissection of the P500 promoter revealed a CPE essential for initiating *PAX6* expression. Thus, *PAX6* expression during NPC fate specification is presumably controlled by G2-specific regulators (Fig. 7). Bioinformatic predictions and candidate approaches  have so far failed to identify TFs driving this process. We note that, while it is relatively straightforward to identify functional cis-regulatory elements in gene regulation, it is notoriously difficult to identify TFs that act through these elements. In addition to *PAX6*, several other genes initiate their expression in G2 during neural induction. It will be interesting to test whether their expression is activated by a similar mechanism.

During human embryogenesis, PAX6 orchestrates neural plate patterning and maintains the undifferentiated state of neuroepithelial cells (Zhang et al, 2010). Symmetric divisions of PAX6+ progenitors ensure rapid expansion of the progenitor pool prior to cortical specialization. Our study reveals that *PAX6* transcription is activated in G2, with subsequent mitosis producing two PAX6+

daughter cells. This G2-specific activation may facilitate symmetric divisions, enabling efficient progenitor pool expansion—a process critical for proper CNS development.

Prior studies have emphasized the G1 phase as a pivotal window for fate regulation, showing correlations between G1 lengthening and differentiation onset (Blomen and Boonstra, 2007; Dalton, 2015; Lange et al, 2009; Pauklin and Vallier, 2013; Salomoni and Calegari, 2010). By combining live-cell imaging, FUCCI reporters, and scRNA-seq, we instead pinpointed *PAX6* activation to the G2 phase. We speculate that G2-phase fate transitions confer certain advantages: chromatin reorganization and transcriptional reprogramming in G1/S phases risk errors during DNA replication, whereas G2-phase commitment immediately precedes mitosis, ensuring faithful inheritance of fate-determining factors by daughter cells (Fig. 7). This mechanism may be particularly beneficial during early development when the expansion of progenitor pools is critical. It will be interesting to examine whether G2-coupled fate commitment occurs in other cell lineages that undergo rapid expansion during development.

Finally, we have shown that S-phase arrest by hydroxyurea (HU) disrupts NPC fate transition and activates mesodermal genes, underscoring the role of the G2 phase in neural lineage specification (Fig. 7). We emphasize that the emergence of mesenchymal signatures after HU treatment only indicates impaired cell fate decisions after cell cycle arrest. This does not normally occur under physiological conditions in vivo. Our results seemingly contradict a recent study by Kukreja *et al*, which reported that zebrafish embryos treated with HU and aphidicolin completed germ layer differentiation despite the cell cycle blockade (Kukreja et al, 2024). This apparent contradiction may reflect differences between species or stem from system-specific compensatory mechanisms. In zebrafish, signaling networks (e.g., Notch and Wnt) may buffer fate determination (Chidiac and Angers, 2023; Zamfirescu et al, 2022), whereas in vitro models lack such feedback loops, heightening sensitivity to cell cycle perturbations. Furthermore, HU treatment timing—aligned with early stage of gastrulation in zebrafish versus early neural induction in our system—may differentially impact fate commitment. Future studies should explore how cell-cell communications and microenvironmental cues modulate the coupling between the cell cycle and cell fate decisions across species.

## Conclusion

Using PAX6 as a fate transition marker in an in vitro differentiation system, we provide the first direct evidence that early neural commitment of human ESCs occurs during the G2 phase. We identify

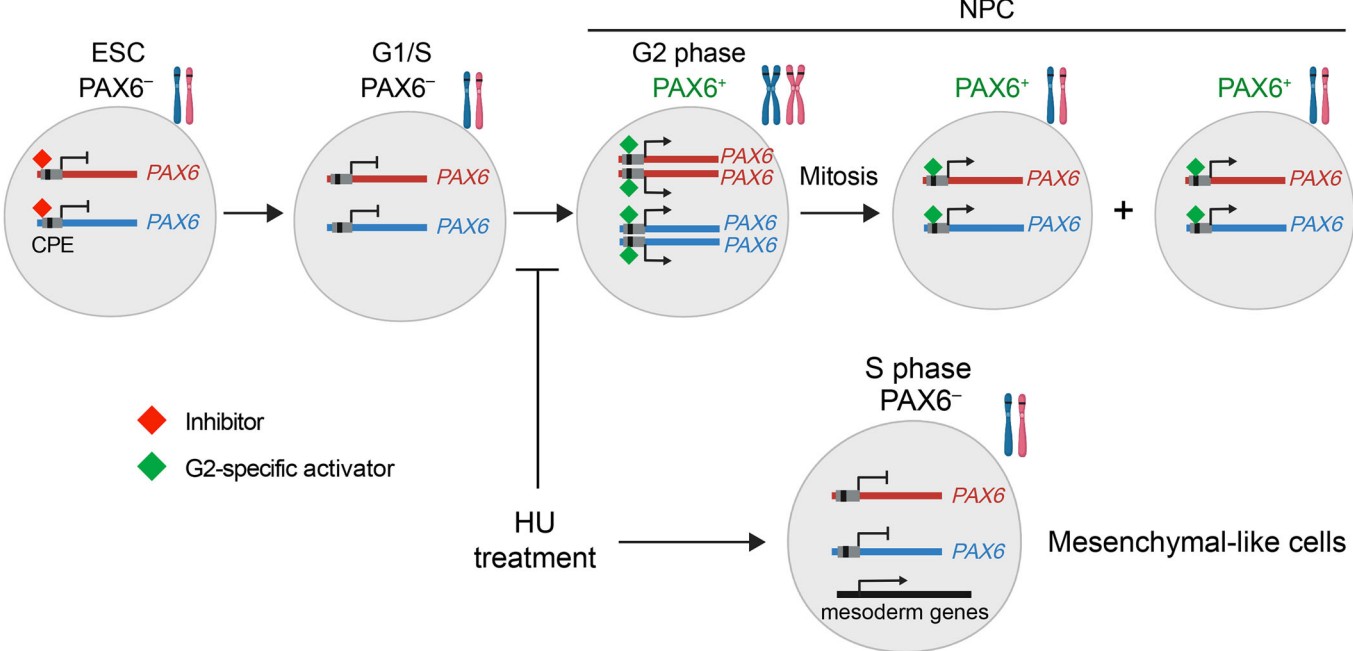

**Figure 7.    Model of G2-coupled *PAX6* activation during ESC–NPC transition.**

*PAX6* transcription is controlled by both negative and positive regulators. The inhibitors exist in ESCs. G2-specific activators that bind the CPE promote *PAX6* expression at G2. This G2-coupled activation of *PAX6* expression allows concerted activation of all four copies of the *PAX6* gene and generation of two daughter cells of identical fate. Blocking the cell cycle at S phase by hydroxyurea (HU) during differentiation prevents PAX6 activation and leads to aberrant expression of mesoderm genes.

a 500-bp *PAX6* promoter governing its transcriptional dynamics and define a core promoter element essential for activation. We further show that S-phase arrest blocks NPC specification by diverting cells toward mesodermal lineages, implicating an instructive role of the cell cycle in fate determination. These findings advance our understanding of how cell cycle dynamics integrate with transcriptional programs to orchestrate lineage transitions during development.

## Methods

### Reagents and tools table

| Reagent/resource | Reference or source | Identifier or catalog number |
| --- | --- | --- |
| **Experimental models** | | |
| H9 embryonic stem cell | WiCell Research Institute | WA09 |
| **Recombinant DNA** | | |
| Plasmid FUCCI | Addgene | Cat #86849 |
| Plasmid LentiCRISPR v2 | Addgene | Cat #52961 |
| Plasmid pMD2.G | Addgene | Cat #12259 |
| Plasmid psPAX2 | Addgene | Cat #12260 |
| Plasmid *PAX6* promoter-driven miRFP680 | This study | / |

| Reagent/resource | Reference or source | Identifier or catalog number |
| --- | --- | --- |
| **Antibodies** | | |
| Rabbit anti-PAX6 | Cell Signaling Technology | Cat #60433 |
| Mouse anti-GAPDH | Proteintech | Cat #60004-1-Ig |
| Mouse anti-OCT3/4 | Santa Cruz Biotechnology | Cat #sc-5279 |
| Goat anti-NANOG | R&D Systems | Cat AF1997 |
| Mouse anti-TUJ1 | R&D Systems | Cat #MAB1195 |
| Rabbit anti-SOX2 | Cell Signaling Technology | Cat #3579S |
| Rabbit anti-PAX6 | Invitrogen | Cat #42-6600 |
| Alexa Fluor® 647 Mouse anti-Oct3/4 | BD Biosciences | Cat #560307 |
| Donkey anti-rabbit IgG (H + L) antibody Alexa Fluor™ 488 | Invitrogen | Cat #A32790 |
| Donkey anti-rabbit IgG (H + L) antibody with Alexa Fluor™ 647 | Invitrogen | Cat #A32795 |
| Donkey anti-rabbit IgG (H + L) antibody with Alexa Fluor™ 594 | Invitrogen | Cat #A32754 |
| Goat Anti-Rabbit IgG H&L | Vazyme | Cat #Ab206-01 |

| Reagent/resource | Reference or source | Identifier or catalog number |
|---|---|---|
| Donkey anti-rabbit IgG (H + L) Dylight 680 conjugates | Cell Signaling Technology | Cat #5366S |
| Donkey anti-mouse IgG (H + L) Dylight 800 conjugates | Cell Signaling Technology | Cat #5257S |
| **Oligonucleotides and other sequence-based reagents** | | |
| *PAX6* qPCR forward primer | This study | CAGAGCCCCATATTCGAGCC |
| *PAX6* qPCR reverse primer | This study | CAAAGACACCACCGAGCTGA |
| *GAPDH* qPCR forward primer | This study | ACACCCACTCCTCCACCTTTG |
| *GAPDH* qPCR reverse primer | This study | TCCACCACCCTGTTGCTGTAG |
| *PAX6* sgRNA#1 | This study | TGGTTGGTATCCGGGGACTT |
| *PAX6* sgRNA#2 | This study | GAGACAGATTACTGTCCGAG |
| *SIX3* sgRNA#1 | This study | ACATGGACAACTCTTCCGGG |
| *SIX3* sgRNA#2 | This study | ATACTTCTGGCGAGTAGCGG |
| *FOXM1* sgRNA#1 | This study | CGGGTCAGCTCATACCTGG |
| *FOXM1* sgRNA#2 | This study | CTACTTGACATTGGACCAGG |
| *LIN54* sgRNA#1 | This study | TAGTTACCATTGGAGGGAGG |
| *LIN54* sgRNA#2 | This study | GTTCTTTTCACAGTCTACTCC |
| *EGR1* sgRNA | This study | ACGCCCTTACGCTTGCCCAG |
| 43 bp deletion sgRNA#1 (left) | This study | CGGCTGGGATGGGGTGGGGG |
| 43 bp deletion sgRNA#2 (right) | This study | TTGCCAACTCCAGGACAGCG |
| *AAVS1* sgRNA#1 | This study | GGGGCCACTAGGGACAGGAT |
| **Chemicals, enzymes and other reagents** | | |
| Matrigel | Corning | Cat #354277 |
| mTeSR1 media | STEMCELL Technologies | Cat #85850 |
| Versene | Thermo Fisher Scientific | Cat #15040066 |
| Y-27632 | STEMCELL Technologies | Cat #72308 |
| MycoBlue Mycoplasma Detector | Vazyme | Cat #D101-02 |
| STEMdiff™ SMADi Neural Induction Kit | STEMCELL Technologies | Cat #08581 |
| Accutase | STEMCELL Technologies | Cat #07920 |
| Hydroxyurea | Selleck | Cat #S1896 |
| STEMdiff Forebrain Neuron Differentiation medium | STEMCELL Technologies | Cat #08600 |
| STEMdiff Forebrain Neuron Maturation medium | STEMCELL Technologies | Cat #08605 |
| ClonExpress MultiS One Step Cloning Kit | Vazyme | Cat #C113-01 |

| Reagent/resource | Reference or source | Identifier or catalog number |
|---|---|---|
| Lipofectamine 2000 | Life Technologies | Cat #11668019 |
| Lenti-X concentrator | Takara | Cat #PT4421-2 |
| FastPure Cell/Tissue Total RNA Isolation Kit V2 | Vazyme | Cat #RC112-01 |
| Guide-it long ssDNA kit | Takara | Cat #632667 |
| iTaq™ Universal SYBR® Green Supermix | Bio-Rad | Cat #1725124 |
| BD Phosflow™ Fix Buffer I | BD Biosciences | Cat #554655 |
| BD Phosflow™ Perm Buffer III | BD Biosciences | Cat #558050 |
| NEBNext Ultra RNA Library Prep Kit for Illumina | New England Biolabs | Cat #E7530L |
| ATAC-Seq Kit | Active Motif | Cat #53150 |
| Hyperactive Universal CUT&Tag Assay Kit for Illumina | Vazyme | Cat #TD903-02 |
| TruePrep DNA Library Prep Kit V2 for Illumina | Vazyme | Cat #TD501 |
| Chromium Next GEM Single Cell Multiome ATAC + Gene Expression Reagent Kits | 10×Genomics | Cat #PN-1000280 |
| T4 Polynucleotide Kinase | New England Biolabs | Cat M0201L |
| T4 DNA Ligase | New England Biolabs | Cat M0202L |
| **Software** | | |
| Prism9 | GraphPad | https://www.graphpad.com/scientific-software/prism/ |
| Adobe Illustrator 2020 | Adobe | https://www.adobe.com/products/illustrator.html |
| Fiji | ImageJ | https://imagej.net/software/fiji/ |
| CytExpert | Beckman Coulter | https://www.mybeckman.cn/flow-cytometry/research-flow-cytometers/cytoflex/software |
| **Other** | | |
| Deltavision microscope | GE Healthcare | N/A |
| Odyssey infrared imaging system | LI-COR Biosciences | N/A |
| Bio-Rad TC20 automated cell counter | Bio-Rad | N/A |
| Beckman Coulter CytoFLEX flow cytometer | Beckman Coulter Life Sciences | N/A |
| Bio-Rad CFX96 | Bio-Rad | N/A |
| LSM 900 confocal microscope | ZEISS | N/A |
| Illumina Novaseq 6000 | Illumina | N/A |

## Human embryonic stem cell (ESC) culture

Human embryonic stem cell line (ESC; H9/WA09) was obtained from WiCell Research Institute, Wisconsin, USA. ESCs were cultured in mTeSR1 media (STEMCELL Technologies, #85850) in six-well plates coated with Matrigel (Corning, #354277) (Ludwig et al, 2006). ESCs were passaged to new plates when they grew to ~70% confluency. The Versene reagent (Thermo Fisher Scientific, #15040066) was used for routine passage, and 10 μM Y-27632 (STEMCELL Technologies, #72308) was added for 24 h to prevent cell apoptosis. All ESCs were cultured at 37 °C in a humidified atmosphere supplemented with 5% $CO_2$. The pluripotency of ESCs was confirmed by immunofluorescence staining of pluripotency markers, such as OCT4, SOX2, and NANOG. ESCs were regularly tested for mycoplasma contamination by using a commercial kit (MycoBlue Mycoplasma Detector, Vazyme, #D101-02).

## Neural progenitor cell (NPC) differentiation and hydroxyurea treatment

ESCs were differentiated into NPCs using a monolayer protocol with the STEMdiff™ SMADi Neural Induction Kit (STEMCELL Technologies, #08581) according to the manufacturer's instructions(Chambers et al, 2009). Briefly, ESCs cultured in mTeSR1 were detached into single-cell suspension using Accutase (STEMCELL Technologies, #07920), and cells were resuspended in the neural induction medium containing SMAD inhibitor supplements. In all, $2 \times 10^6$ cells were seeded into Matrigel-coated six-well plate in the neural induction medium with SMAD inhibitors, supplemented with 10 μM Y-27632. The cells were replenished with fresh neural induction medium containing SMAD inhibitors every day and passaged to new plates every 6 days. 10 μM Y-27632 was added for 24 h to prevent cell apoptosis after each cell passaging.

For cell cycle synchronization at the S phase, 250 μM hydroxyurea (HU) was added to the neural induction medium from day 1 (D1) or D2 of differentiation, and these cells were cultured for another 48 or 24 h, respectively, until D3 of differentiation before analysis of PAX6 expression.

## Forebrain neuron differentiation

For forebrain-type neuron (FBN) differentiation, the ESC-derived mature NPCs (passage 3; D18) were detached using Accutase and seeded onto a Matrigel-coated six-well plate at a density of $1.25 \times 10^5$ cells/cm² in STEMdiff Neural Induction Medium containing SMAD inhibitors for 24 h. The medium was then changed to STEMdiff Forebrain Neuron Differentiation medium (STEMCELL Technologies, #08600) and replenished with fresh medium daily for 6 days. On D7, forebrain neural precursors were detached using Accutase and seeded onto a Matrigel-coated six-well plate at a density of $5 \times 10^4$ cells/cm² in STEMdiff Forebrain Neuron Maturation medium (STEMCELL Technologies, #08605) (Kuroda et al, 2024), with a full-medium change every 2 d for 15 d.

## Plasmids

For gene editing, the sgRNAs targeting the gene of interest were subcloned into the pLenti-CRISPR v2 vector (Addgene, #52961), and the sgRNA targeting the *AAVS1* site was used as a negative control. For the *PAX6* reporter assay, the promoters of *PAX6* with different lengths were amplified from the genomic DNA of ESCs and ligated with cDNAs encoding the miRFP680 reporter in a lentiviral plasmid backbone. The ClonExpress MultiS One Step Cloning Kit (Vazyme, #C113-01) and Mut Express II Fast Mutagenesis Kit V2 (Vazyme, #C214-01) were used to generate these constructs.

## Lentivirus packaging and generation of FUCCI ESCs

The lentiviral targeting vectors were transfected into HEK293FT cells along with psPAX2 (Addgene, #12260) and pMD2.G (Addgene, #12259) using Lipofectamine 2000 (Life Technologies, #11668019). Following transfection, viral supernatant was collected at 2–3 days and concentrated using a Lenti-X concentrator (Takara, #PT4421-2).

The FUCCI vector was obtained from Addgene (#86849) and used for generating lentiviruses. ESCs were infected with the FUCCI lentivirus and then treated with 0.5 μg/ml puromycin for 3 days. The surviving cells were picked as single clones to establish the FUCCI-ESC cell line. FUCCI signals and pluripotency markers of the FUCCI ESCs were verified by immunofluorescence staining.

## Western blotting

The cell pellets were lysed in 1×RIPA buffer containing 1 mM PMSF protease inhibitor and genomic DNA was sheared by sonication. The resulting lysates were cleared through centrifugation at 4 °C, analyzed by SDS-PAGE, and then transferred to the PVDF membrane using the wet-transfer protocol. The primary antibodies used in this study included: rabbit anti-PAX6 (Cell Signaling Technology, #60433); mouse anti-GAPDH (Proteintech, #60004-1-Ig); mouse anti-OCT3/4 (Santa Cruz Biotechnology, #sc-5279). The secondary antibodies used were donkey anti-rabbit IgG (H + L) Dylight 680 conjugates (Cell Signaling Technology, #5366S) and donkey anti-mouse IgG (H + L) Dylight 800 conjugates (Cell Signaling Technology, #5257S). The membranes were scanned and quantified using the Odyssey infrared imaging system (LI-COR Biosciences).

## Quantitative real-time PCR

Total RNAs were extracted from ESCs, NPCs, or FBNs using the FastPure Cell/Tissue Total RNA Isolation Kit V2 (Vazyme, #RC112-01), reverse transcribed and amplified to prepare the long ssDNA using the Guide-it long ssDNA kit (Takara, #632667) following the manufacturer's instructions. Real-time qPCR was performed by using iTaq™ Universal SYBR® Green Supermix (Bio-Rad, #1725124) in a Bio-Rad CFX96 system.

## Immunofluorescence staining

ESCs, NPCs, and FBNs were cultured on Matrigel-coated cover glasses (Thermo Fisher Scientific, #1254580). The cells were briefly rinsed with 1×PBS, then fixed in 4% paraformaldehyde solution for 15 min at room temperature. The cells were permeabilized in PBS containing 0.2% Triton for 10 min and then incubated in the blocking buffer consisting of 3% BSA and 0.1% Tween 20 in PBS for 1 h at room temperature. After blocking, the cells were incubated with primary antibodies diluted in the

blocking buffer for 2 h at room temperature. The cells were subsequently washed three times with the wash buffer consisting of 0.1% Tween 20 in PBS, then incubated with secondary antibodies diluted in the blocking buffer for 1 h at room temperature. After another three washes in the wash buffer, the cells were co-stained with 1 µg/ml DAPI. The cover glasses were then mounted on glass slides using the Fluoromount-G® Mounting Medium (SouthernBiotech, #0100-01) and sealed with nail polish. Slides were imaged using a 40×oil/water objective on a ZEISS 900 confocal microscope. ImageJ was used for image processing and quantification (Schneider et al, 2012).

The primary or secondary antibodies used for staining included rabbit anti-PAX6 (Cell Signaling Technology, #60433), mouse anti-OCT3/4 (BD Biosciences, #560307), goat anti-NANOG (R&D Systems, AF1997), mouse anti-TUJ1 (R&D Systems, #MAB1195), donkey anti-rabbit IgG (H + L) antibody Alexa Fluor™ 488 (Invitrogen, #A32790), donkey anti-rabbit IgG (H + L) antibody with Alexa Fluor™ 647 (Invitrogen, #A32795), and donkey anti-rabbit IgG (H + L) antibody with Alexa Fluor™ 594 (Invitrogen, #A32754).

## Flow cytometry

The cells were dissociated into single cells with Accutase and briefly washed with PBS. The cell pellets were then fixed with BD Phosflow™ Fix Buffer I (BD Biosciences, #554655) for 20 min at room temperature. After two washes with PBS, the cells were resuspended in BD Phosflow™ Perm Buffer III (BD Biosciences, #558050) and incubated for 30 min on ice or stored at −20 °C. For staining with the PAX6 antibody, ~$5 \times 10^5$ cells stored in BD Phosflow™ Perm Buffer III were washed twice with PBS containing 1% BSA and then incubated with the rabbit anti-PAX6 antibody (Invitrogen, #42-6600) at 1:50 dilution for 2 h at room temperature. After two washes with PBS containing 1% BSA, the cells were incubated with Alexa Fluor-488 conjugated donkey anti-rabbit IgG (H + L) antibody (Invitrogen, #A32790) at a 1:1000 dilution for 1 h in the dark at room temperature. After two additional washes with PBS containing 1% BSA, the cells were resuspended in 100 µl of PBS containing 1 µg/ml DAPI. Finally, the cells were filtered using a 40-µm cell strainer and analyzed on a CytoFLEX LX instrument (Beckman).

## Time-lapse imaging

The differentiating cells were cultured in Matrigel-coated Nunc Lab-Tek chambered cover glass (Thermo Fisher Scientific; cat#155411). Imaging was conducted at 5-min intervals over a period of 30–72 h after neural induction, using a ×40 objective on a Delta Vision microscope (GE Healthcare). The microscope was equipped with an environmental chamber regulating temperature and $CO_2$ levels. Time-lapse videos were processed and quantified using ImageJ.

## RNA sequencing (RNA-seq)

All bulk RNA-seq experiments were conducted with biological duplicates. Total RNAs from ESCs, NPCs, or FBNs were extracted using the FastPure Cell/Tissue Total RNA Isolation Kit V2 (Vazyme, #RC112-01), and sequencing libraries were generated using the NEBNext Ultra RNA Library Prep Kit for Illumina (NEB, USA,

#E7530L) following the manufacturer's instructions. Briefly, mRNAs were purified from total RNAs using poly-T oligo-attached magnetic beads. Fragmentation was conducted using divalent cations at an elevated temperature in NEB Next First Strand Synthesis Reaction Buffer (5×). First-strand cDNAs were synthesized using random hexamer primers and M-MuLV reverse transcriptase (RNase H). Subsequently, second-strand cDNA synthesis was performed using DNA polymerase I and RNase H. The remaining overhangs were converted into blunt ends through exonuclease/polymerase activities. Following adenylation of the 3' ends of DNA fragments, NEB Next Adaptors with hairpin loop structures were ligated for hybridization preparation. cDNA fragments within the desired length range of 370 ~ 420 bp were purified using the AMPure XP system (Beverly, USA). 3 µl USER Enzyme (NEB, USA) was incubated with size-selected, adaptor-ligated cDNA at 37°C for 15 min, followed by 5 min at 95°C prior to PCR. PCR was performed using the Phusion high-fidelity DNA polymerase, universal PCR primers, and the index (X) primer. The PCR products were purified using the AMPure XP system, and the library quality was assessed on the Agilent 5400 System (Agilent, USA). The libraries were then quantified using QPCR (1.5 nM) and pooled for sequencing with the PE150 strategy on Illumina platforms from Novogene Bioinformatics Technology Co., Ltd (Beijing, China), based on the effective library concentration and desired data amount.

The quality control of FASTQ files was conducted using FastQC (v0.11.9) and Fastp (Chen et al, 2018). The sequences were aligned to the Hg38 human genome build using the STAR alignment tool (v2.7.8a) (Dobin et al, 2013). The quantification of gene expression was performed using the raw count metrics, based on RefSeq gene annotation, with the aid of FeatureCounts in subread (v2.0.2) (Liao et al, 2014). Differential expression analysis was performed using the "DESeq2" (v1.38.3) (Love et al, 2014) package in R. Data visualization was done using the "ggplot2" package in R.

## 4D label-free quantitative proteomics

For protein extraction, the cell samples were ground into powder in liquid nitrogen and transferred to a 5-ml centrifuge tube. Subsequently, 4 volumes of the lysis buffer consisting of 8 M urea and 1× protease inhibitor cocktail were added to the cell powder. The mixture was subjected to sonication for 3 min on ice using a high-intensity ultrasonic processor (Scientz). The resulting solution was centrifuged at $12,000 \times g$ at 4 °C for 10 min to remove any remaining debris. The supernatant was collected, and the protein concentration was determined using the BCA kit following the manufacturer's instructions.

For the digestion process, the protein solution was incubated in 5 mM dithiothreitol for 30 min at 56 °C. Subsequently, alkylation was performed by introducing 11 mM iodoacetamide and reacting for 15 min at room temperature in the dark. The protein samples were diluted with 100 mM TEAB to decrease the urea concentration to less than 2 M. Trypsin was added to the diluted samples at a trypsin-to-protein mass ratio of 1:50 for the initial digestion overnight, followed by a second 4-h digestion at a trypsin-to-protein mass ratio of 1:100. The resulting peptides were purified and separated from impurities using a C18 solid-phase extraction (SPE) column.

For LC-MS/MS analysis, the tryptic peptides were dissolved in solvent A (a mixture of 0.1% formic acid and 2% acetonitrile in water) and directly loaded onto a homemade reversed-phase

analytical column, which had a length of 25 cm and an inner diameter of 100 μm. The mobile phase consisted of solvent A and solvent B (0.1% formic acid in acetonitrile). The peptides were separated using the following gradient: 0-70 min with a linear increase from 9% to 30% of solvent B, 70–82 min with a linear increase from 30% to 40% of solvent B, 82–86 min with a linear increase from 40% to 90% of solvent B, and 86–90 min at a constant concentration of 90% of solvent B. The separation was performed at a constant flow rate of 450 nl/min using an Easy-nLC1000_TOF UHPLC system (Bruker Daltonics).

The MS/MS data obtained were processed using the MaxQuant search engine (version 1.6.15.0). Tandem mass spectra were searched against the Homo_sapiens_9606_SP_20230103.fasta database, which contains 20389 entries, and was concatenated with a reverse decoy and contaminants database. Trypsin/P was specified as the cleavage enzyme, allowing up to two missing cleavages. The minimum peptide length was set to seven amino acids, and the maximum number of modifications per peptide was set to 5. The mass tolerance for precursor ions was set to 20 ppm for both the first search and main search, and the mass tolerance for fragment ions was also set to 20 ppm. Carbamidomethyl on cysteine was specified as a fixed modification, while acetylation on protein N-terminal and oxidation on methionine were specified as variable modifications. To ensure high confidence in the identified results, false discovery rate (FDR) thresholds were adjusted to achieve an FDR of less than 1% for proteins, peptides, and peptide-spectrum matches (PSMs).

Fisher's exact test was used to determine the significance of the functional enrichment of differentially expressed proteins, with the identified protein as the background. Significant functional terms were identified based on a fold enrichment (log2FC) greater than 0.5 and a $P$ value less than 0.05.

## Time-series cluster (Mfuzz) analysis

For identification of proteins with significant changes during neural induction, the relative expression levels of proteins were first transformed into logarithmic values using a Log2 transformation, and then protein candidates with SD > 0.5 were selected. For the MFuzz analysis, the clustering number (k) was set to 4, and the fuzziness coefficient (m) was set to 2. Visualization of cluster results were performed using the mfuzz.plot2 function in the Mfuzz package (Kumar and Futschik, 2007).

## ATAC-seq

All ATAC-seq experiments were conducted in biological duplicates using the commercial kit (Active Motif, #53150). Briefly, 100,000 cells were used to isolate nuclei. Following cell lysis in an ice-cold ATAC lysis Buffer, nuclei were incubated in the tagmentation master mix for 30 min at 37 °C, and the resulting DNA was purified using DNA purification columns. PCR amplification of the tagmented DNA was then performed to generate libraries with appropriately indexed primers. Following SPRI bead clean-up, the DNA libraries were sequenced on the Illumina NovaSeq platform (PE-150, 50 million reads) at Hangzhou Repugene Technology Co., Ltd.

The quality control of the ATAC-seq FASTQ files was conducted using Cutadapt (v4.0) (Martin, 2011). The sequencing data were aligned to the hg38 human genome assembly using BWA. Duplicate reads were removed using Picard, and only non-duplicate

reads in the BAM format were used for subsequent analysis. Peak calling on the nucleosome-free reads was performed with MACS2(v2.2.9) (Zhang et al, 2008) using the 'callpeak' module and the following parameters: –shift -100 –extsize 200. For data normalization and visualization, the BAM files were converted to the bigWig format using "bamCoverage" with CPM in deepTools (v3.5.3)(Ramírez et al, 2014). Heatmaps and average profiles were generated using the "plotHeatmap" and "plotProfile" scripts in deepTools. To annotate the location of the ATAC-Seq peaks in terms of important genomic features, their BED files were assigned to promoters (defined as -2 kb from the transcription start site), introns, intergenic regions, exons, etc. using the "ChIPseeker" package in R (v1.42.1) (Yu et al, 2015). Using the "DiffBind" package in R (v3.8.4), peaks with significant differences between groups were selected by applying two criteria: $P$ value < 0.05 and absolute fold change (FC) > 2, and gene region annotation was performed using the "ChIPseeker" package in R.

## Cleavage under targets and tagmentation (CUT&Tag)

Approximately $1 \times 10^5$ ESCs, NPC_D2 cells, or NPC_D3 cells were collected and analyzed using the Hyperactive Universal CUT&Tag Assay Kit for Illumina (Vazyme, TD903-02), following the manufacturer's instructions. Each sample included two biological replicates. Briefly, fresh cell cultures were harvested and counted to obtain $1 \times 10^5$ cells per sample. Cells were then incubated with Concanavalin A-coated beads at room temperature for 10 min. After removal of the supernatant, cells were resuspended in ice-cold Antibody Buffer containing the PAX6 antibody (Cell Signaling Technology, #60433) and incubated overnight at 4 °C. The primary antibody solution was then removed, and the cells were incubated with the secondary Goat Anti-Rabbit IgG H&L (Vazyme, Ab206-01) diluted in DIG Wash Buffer for 1 h at room temperature. After thorough washing, CUT&Tag pA/G-Tn5 Transposomes were added and incubated for 1 h at room temperature. Following additional washes, tagmentation was performed at 37 °C for 1 h. DNA was extracted using magnetic DNA purification beads. The purified DNA was amplified via PCR using indexed i7 and i5 primers (Vazyme, TruePrep DNA Library Prep Kit V2 for Illumina, #TD501). The resulting libraries were cleaned, quantified, and subjected to Illumina sequencing on the NovaSeq platform (PE-150, 50 million reads) at Hangzhou Repugene Technology Co., Ltd.

CUT&Tag FASTQ files were mapped to hg38 reference genome with Bowtie2 (Langmead and Salzberg, 2012) and SAM-format mapping files were sorted, quality-checked, and format-transformed with SAMtools command "sort", "flagstat" and "view" (Danecek et al, 2021). Software and pipelines for duplicate reads exclusion, format transformation, peak calling and significantly different peak analysis were performed as described for ATAC-seq data. AnnotationHub (v3.6.0), "ChIPseeker" (v1.42.1) (Wang et al, 2022) and GenomicRanges (v1.50.2) (Lawrence et al, 2013) in R were used to annotate target regions in the genome with peaks. Visualization of interested peaks was generated by IGV (Robinson et al, 2011).

## Single-cell multiome ATAC+RNA sequencing

The single-cell (sc) RNA-seq and ATAC-seq were performed using a 10×Genomics kit (PN-1000280). Cells were collected at different timepoints, counted using the Countess II FL Automated Cell

Counter (Thermo Fisher Scientific), and then lysed on ice for 5 min. Subsequently, 12,000 isolated nuclei were transposed and loaded onto the Chromium Next GEM Chip J (10×Genomics). The chip containing the cells was then loaded onto a Chromium Controller, and the library construction was performed according to the manufacturer's instructions. Sequencing of the gene expression library was performed on an Illumina Novaseq 6000 platform with the PE150 reading strategy. The scATAC library was sequenced on the Illumina Novaseq 6000 platform with a sequencing depth of at least 25 k read pairs per nucleus with the PE50 reading strategy. Sequencing was performed by Hangzhou Lianchuan Biology Technology Co. Ltd., China.

For the preprocessing of single-cell multiomics data, the raw sequencing data resulting from single-cell multiomics experiments were processed using Cell Ranger ARC 1.0.1 software to perform various analyses related to gene expression and chromatin accessibility. Reference human genome data (GRCh38) for Cell Ranger ARC was downloaded from https://cf.10xgenomics.com/supp/cell-arc/refdata-cellranger-arc-GRCh38-2020-A-2.0.0.tar.gz. Finally, cellranger-arc was used to generate feature counts at the single-cell level for scRNA-seq/ATAC-seq analysis.

For merging multiple single-cell RNA-seq counts matrices from different samples generated by Cell Ranger ARC, the merge function of Seurat (v5.0) in R (4.2.1) was used (Zhao et al, 2021). Low-quality cells with few genes, redundant mitochondrial genes in scRNA-seq dataset or extreme nucleosome signals, low transcription start site (TSS) enrichment score, low fragment number and percentage of reads within peak regions in scATAC-seq were filtered out, resulting in the retention of 70,887 high-quality cells for downstream analysis.

Downstream analysis of RNA-seq was performed with Seurat (v5.0) in R (v4.2.1). Dimensionality reduction and clustering were performed on all cells. Principal component analysis (PCA) and UMAP algorithms were applied to check clustering of samples from each timepoints. Distributions of NPC markers, including PAX6, among cell clusters were shown by Featureplot in Seurat.

The CellCycleScoring function from the Seurat package was used to assign cell cycle phases to individual cells (Kowalczyk et al, 2015). This function calculates cell cycle signatures by averaging the expression levels of genes associated with specific cell cycle phases, using $\log2(TPM + 1)$ values, based on gene sets previously defined in synchronized HeLa cells (Whitfield et al, 2002). Cells not assigned to S and G2/M phases were annotated as G1 phase. Cell cycle phase and development timepoints were combined to create the more delicate phase-level timepoint labels and assigned to each cell. Comparisons of each marker gene expression among different cell cycle phases, development timepoints, or phase-level timepoints were shown by Dotplot in Seurat.

Differential expression gene analysis for cell clusters of phase-level timepoints along the development time axis was detected using FindMarkers in Seurat with the Wilcoxon test (Love et al, 2014). The threshold of differential expression of adjusted $P$ value is 0.05 and that of log2FoldChange is ±2. Volcano plots of differential expression genes were shown using ggplot2 (v3.5.0). GO annotation (Biological Process) of detected differential expression genes was performed with clusterProfiler package (v4.6.2) and the results were shown with dotplot in the enrichplot package (v1.18.4).

Seurat (v5.0) and Signac (v1.12) (Stuart et al, 2021) were used for downstream analysis of single-cell ATAC-seq data. Term frequency inverse document frequency (TF-IDF) normalization was applied on sparse scATAC-seq matrix. Dimension reduction was calculated with partial singular value decomposition (SVD) and UMAP algorithm. Smart Local Moving (SLM) algorithm was applied to perform graph-based clustering in scATAC-seq data. Peak calling on scATAC-seq data was performed with MACS2 (v2.2.9) (Zhang et al, 2008). Chromatin accessibility dynamics on NPC marker genes across development time was visualized by "CoveragePlot()" and "TilePlot()" in Signac.

## Gene ontology analysis

In the analysis of the bulk RNA-seq dataset, up- or downregulated genes were selected based on differential expression analysis in the R package DESeq2 (v1.38.3) (Love et al, 2014) with thresholds of log2FoldChange = 1.5 and adjusted $P$ value = 0.05. For gene ontology analysis, the up- or downregulated genes were input into the enrichGO function in the R package clusterProfiler (v4.6.2) (Wu et al, 2021) with "Biological Process" as the ontology dataset, "org.Hs.eg.db" as the annotation database, and with threshold cutoff = 0.05. The top 10 Gene Ontology categories were chosen for visualization based on $-\log10$(adjusted $P$ value).

Up- or downregulated genes between cell clusters of the scRNA-seq dataset were selected based on the FindMarkers function from Seurat (v5.1.0) with default parameters. GO enrichment in scRNA-seq was performed by compareCluster in R package clusterProfiler (v4.6.2) with similar parameters as those used in bulk RNA-seq except the threshold cutoff was set at 0.01.

For the analysis of ATAC-seq datasets with peak-calling, up- or downregulated peaks were selected by DESeq2 (v1.38.3) and converted into covered genes by the R package GenomicRanges (v1.50.2) and TxDb.Hsapiens.UCSC.hg38.knownGene (v3.16.0) as the input gene list for GO enrichment. GO enrichment of the ATAC-seq and visualization was performed as described for bulk RNA-seq.

## Quantitation and statistical analysis

GraphPad Prism 7 was used to perform the statistical analysis of Western blot, qPCR, confocal imaging, and flow cytometry data. All data were shown as mean ± SD. The Student's unpaired two-tailed $t$ test was applied for significance evaluation between the indicated groups. The one-way analysis of variance (ANOVA) was applied to assess pairwise differences. *$P < 0.01$, **$P < 0.01$, ***$P < 0.001$, ****$P < 0.0001$. The number of biological replicates was indicated in the relevant method sections.

# Data availability

The multiomics datasets generated in this study were submitted to NCBI, including GSE291903, GSE291905, GSE291907, GSE291908, and GSE305658. This study did not generate new codes, and all codes used are available online (https://github.com/Rong-ao/NPC_multiomics.git). Any additional information required to reanalyze the data reported in this paper is available from the lead contact, Hongtao Yu (yuhongtao@westlake.edu.cn), upon request.

The source data of this paper are collected in the following database record: biostudies:S-SCDT-10_1038-S44318-025-00605-y.

## Peer review information

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

## Acknowledgements

We are grateful to Shang Cai, Yihan Wan, and Qi Hu from School of Life Science, Westlake University, for their advice during the project. We also thank Xilai Ding and other members of the FACS core facility at Westlake University for technical support and the Westlake University High-Performance Computing Center for computational resources. This work was supported by the National Natural Science Foundation of China (Project 32130053 to HY) and the New Cornerstone Science Foundation (to HY).

## Author contributions

**Song Hu**: Conceptualization; Resources; Data curation; Formal analysis; Validation; Methodology; Writing—original draft; Writing—review and editing. **Rongao Kou**: Data curation; Formal analysis; Investigation; Visualization; Methodology; Writing—review and editing. **Zhuojie Su**: Data curation; Formal analysis; Investigation; Visualization; Methodology; Writing—review and editing. **Guanchen Li**: Data curation; Formal analysis; Writing—review and editing. **Shutao Qi**: Formal analysis; Investigation; Visualization; Methodology; Writing—original draft; Writing—review and editing. **Yanxiao Zhang**: Formal analysis; Visualization; Methodology; Writing—review and editing. **Haifeng Wang**: Supervision; Writing—review and editing. **Ling-ling Chen**: Supervision; Writing—review and editing. **Hongtao Yu**: Conceptualization; Funding acquisition; Writing—original draft; Project administration; Writing—review and editing.

Source data underlying figure panels in this paper may have individual authorship assigned. Where available, figure panel/source data authorship is listed in the following database record: biostudies:S-SCDT-10_1038-S44318-025-00605-y.

## Disclosure and competing interests statement

The authors declare no competing interests.

# Expanded View Figures

**Figure EV1.  Transcriptomic and Proteomic characterization of cells during ESC–NPC differentiation.**

(A) Volcano plot showing up- and downregulated genes in NPC_D3 compared to NPC_D2. The horizontal and vertical dotted lines indicate *P* value < 0.01 and |log2FC| >1.5, respectively. $n = 2$ independent experiments. (B) Volcano plot showing up- and downregulated proteins in NPC_D3 compared to NPC_D2. The horizontal and vertical dotted lines indicate *P* value < 0.05 and |log2FC| >0.5, respectively. $n = 2$ independent experiments. (C) Top 10 enriched Gene Ontology (GO) terms of upregulated genes in NPC_D3 compared to NPC_D2. (D) Top 10 enriched GO terms of upregulated proteins in NPC_D3 compared to NPC_D2. (E) Scatter plot showing the correlation between the changes in transcripts and the corresponding proteins between NPC_D3 and NPC_D2. The color of the points indicates the density of points at that location. (F) Venn diagram showing the count numbers of up- and downregulated transcripts and proteins in NPC_D3 compared to NPC_D2. (G) Time-series analysis of the proteomics (mfuzz) data of ESCs and NPCs at indicated times of neural induction. (H) GO analysis of the four protein clusters in (G). For the volcano plots, statistical analyses were performed using the negative binomial distribution model for transcriptomic data (A) and the *t* test for proteomic data (B). Statistical significance of GO enrichment analysis in (C, D) is determined using the hypergeometric test and Benjamini-Hochberg false discovery rate (FDR) correction, with a significance threshold of $P < 0.05$.

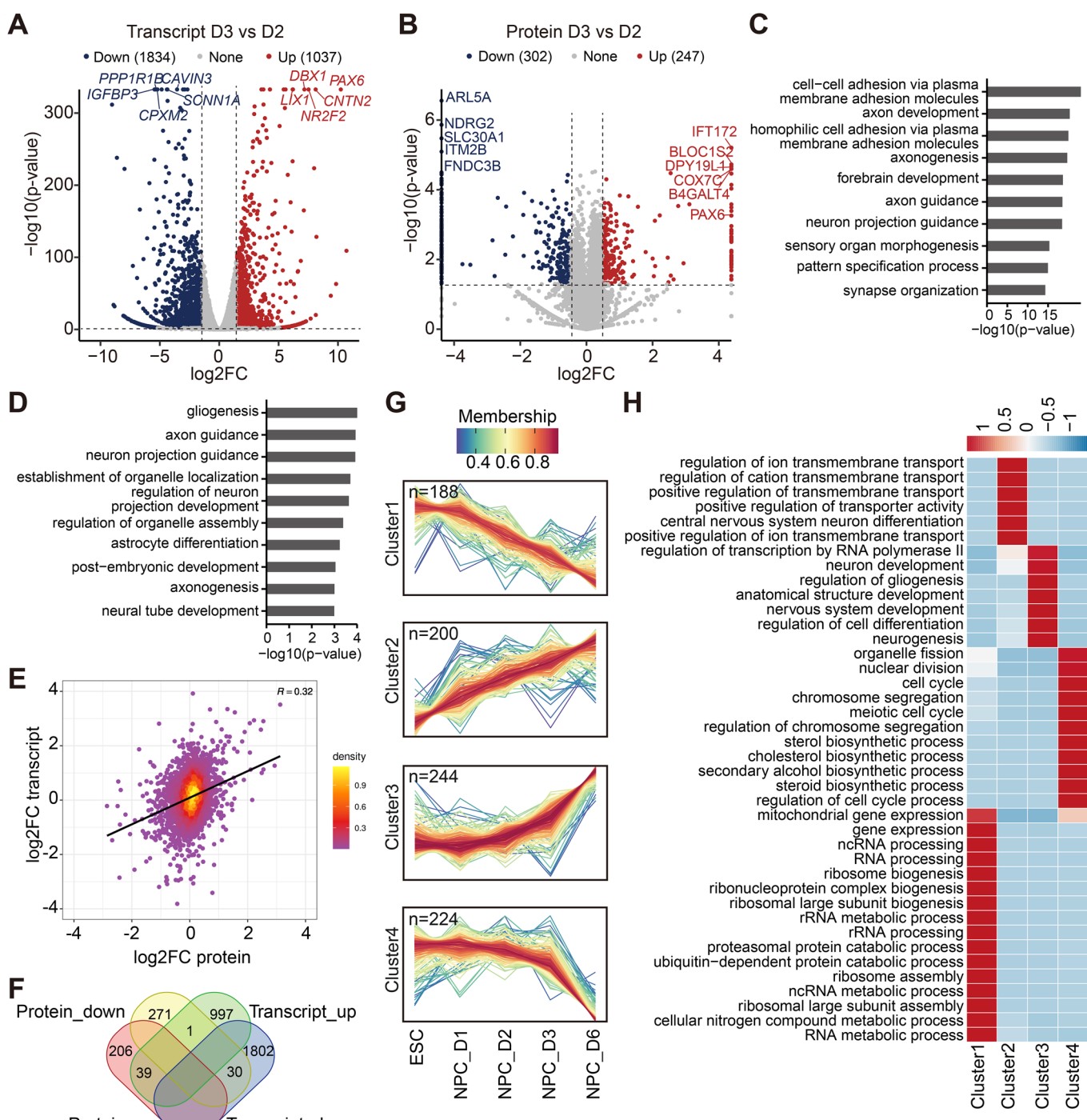

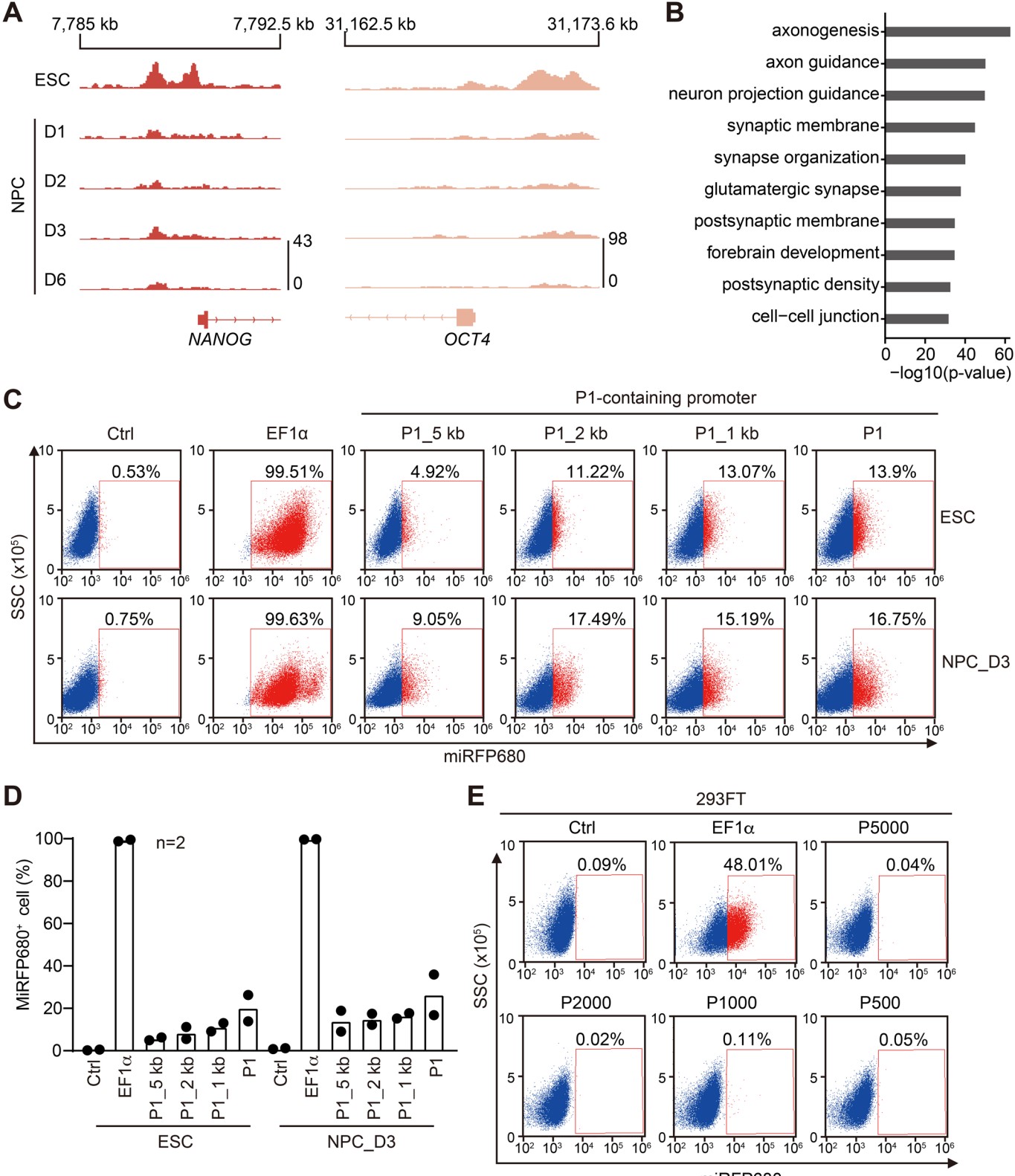

◀

**Figure EV2.   G2-enhanced PAX6 transcription is driven by the 500-bp promoter.**

(A) ATAC-seq tracks showing the chromatin accessibility at the *NANOG* and *OCT4* gene loci in ESCs and NPCs from days 1–6 of neural induction. (B) Top 10 enriched Gene Ontology (GO) pathways of genes with increased chromatin accessibility in NPC_D3 compared to NPC_D2. Statistical significance of GO enrichment analysis is determined using the hypergeometric test and Benjamini-Hochberg false discovery rate (FDR) correction, with a significance threshold of $P < 0.05$. (C) FACS plots showing the expression of the miRFP680 reporter driven by *PAX6* P1-containing promoters of different lengths in ESCs and NPCs at day 3 of neural induction (NPC_D3). The EF1α promoter was used as the positive control. Uninfected cells (Ctrl) were used as the negative control. (D) Quantification of the percentage of miRFP680$^+$ cells in (C). $n = 2$ independent experiments. (E) FACS plots showing the expression of the miRFP680 reporter driven by P500-containing promoters in 293FT cells. The EF1α promoter was used as the positive control, and uninfected cells (Ctrl) were used as the negative control.

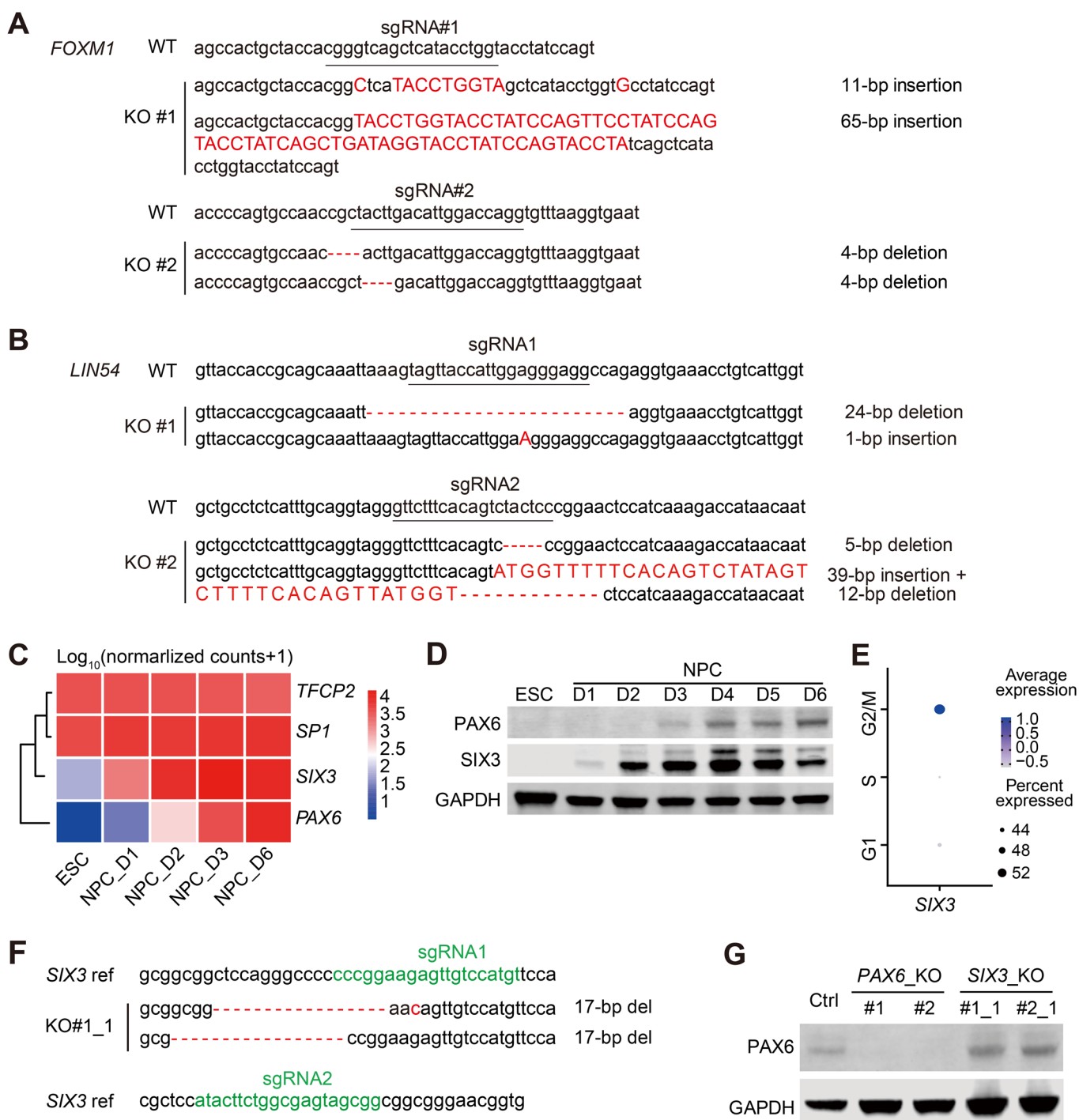

**Figure EV3. FOXM1, LIN54, and SIX3 and EGR1 are not required for *PAX6* expression during ESC–NPC transition.**

(A, B) Sanger sequencing of the *FOXM1* (A) and *LIN54* (B) genes in different ESC knockout (KO) clones. The sgRNA sequences used for the genome editing were underlined in the wild-type (WT) reference sequence. Indels and insertions in the genome were shown in red font. (C) Heatmap showing the expression levels of putative *PAX6* upstream regulators, such as *TFCP2*, *SP1*, and *SIX3* in ESCs and NPCs at different times of neural induction. *PAX6* expression was included as a positive control. The color bar represented values calculated as log10 (normalized counts + 1). (D) Western blots showing the protein levels of SIX3 and PAX6 in ESCs and NPCs. (E) Dot plot showing the expression levels of *SIX3* in different cell cycle phases from NPC_24h to NPC_52h based on the integrated scRNA-seq data. (F) Sanger sequencing of the *SIX3* gene in two ESC KO clones. The sgRNA sequences were highlighted in green, and indels in the genome were indicated in red. (G) Western blots showing the protein levels of PAX6 in WT and *SIX3* KO NPCs on D3 of neural induction. Two *PAX6* KO NPC_D3 were used as the positive control.

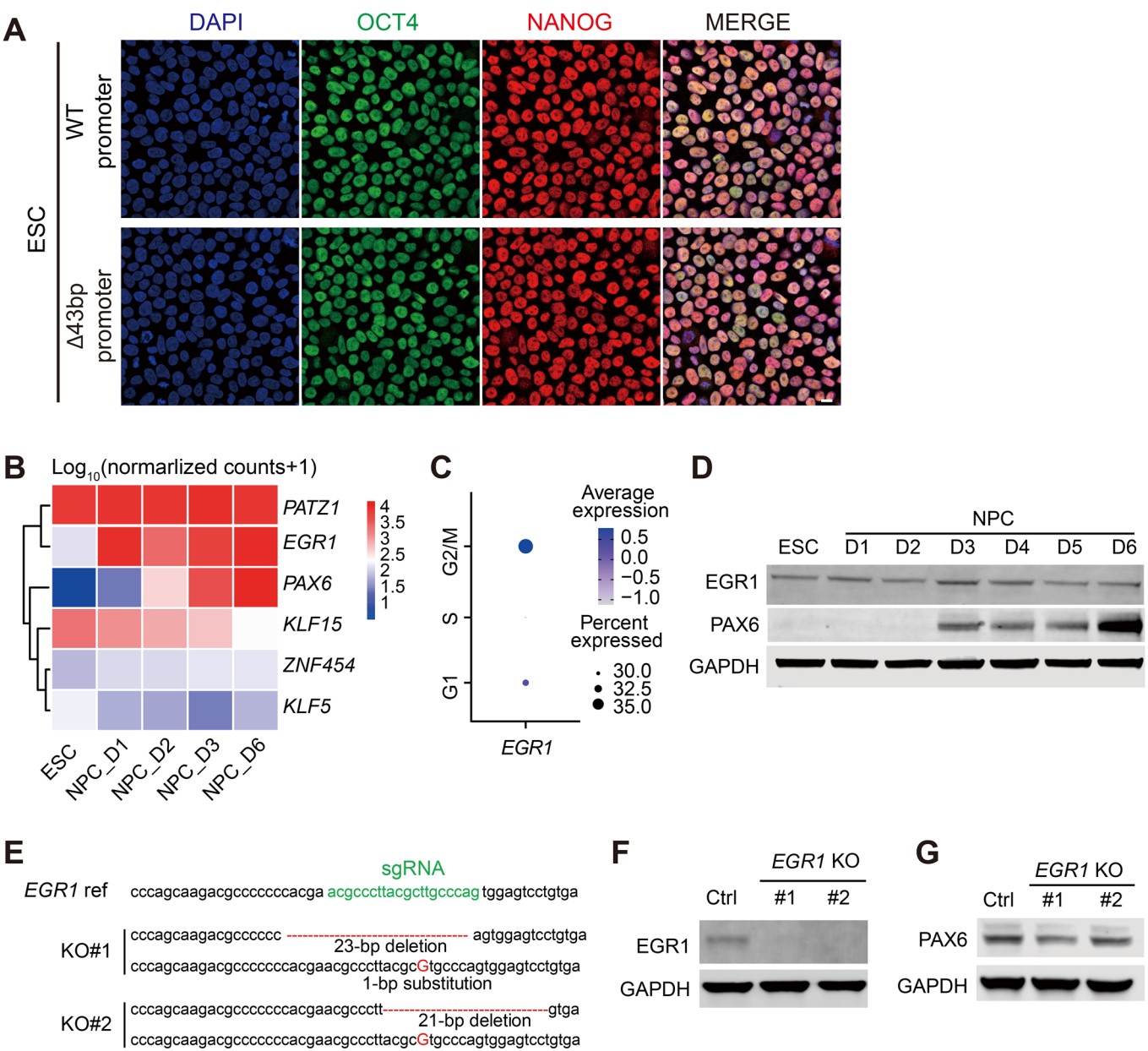

**Figure EV4. SIX3 and EGR1 are not required for *PAX6* expression during ESC–NPC transition.**

(A) Images of WT ESCs or ESCs with the Δ43 bp deletion from the *PAX6* promoter stained with DAPI and antibodies against OCT4 (green) and NANOG (red). Scale bar, 10 μm. (B) Heatmap showing the expression levels of predicted upstream regulators of *PAX6* in ESCs and NPCs at different times of neural induction. (C) Dot plot showing the expression levels of *EGR1* in different cell cycle phases from NPC_24h to NPC_52h based on the integrated scRNA-seq data. (D) Western blots showing the protein levels of EGR1 and PAX6 in ESCs and NPCs. (E) Sanger sequencing of the *EGR1* gene in two ESC knock out (KO) clones. The sgRNA sequence was highlighted in green, and indels in the genome were indicated in red. (F, G) Western blots showing the protein levels of EGR1 in WT and *EGR1* KO ESCs (F) and PAX6 in WT and *EGR1* KO NPCs on D3 of neural induction (G).

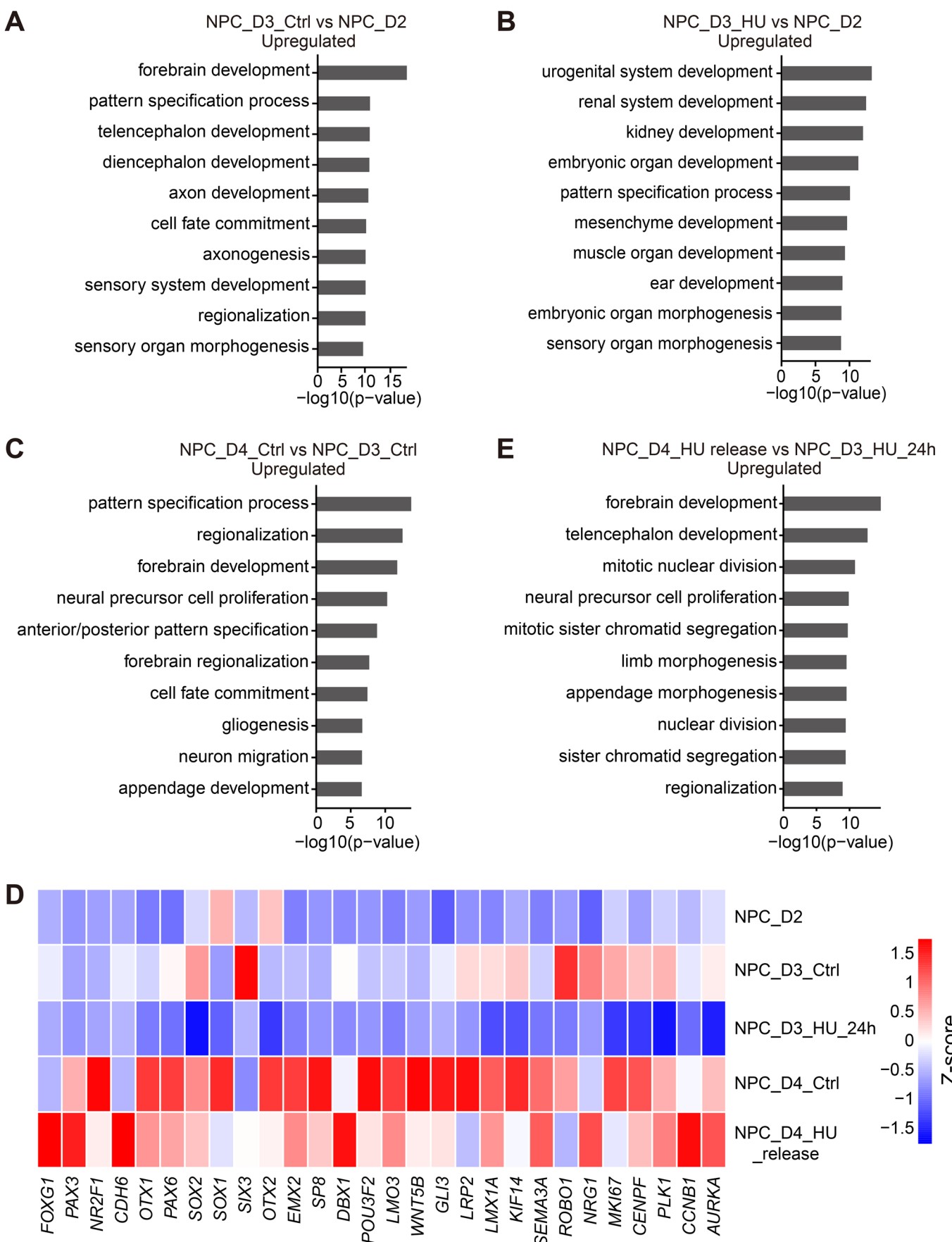

◄ **Figure EV5. Hydroxyurea (HU) treatment impairs the ESC–NPC transition.**

(A) Top 10 enriched Gene Ontology (GO) pathways of upregulated genes in normally differentiated NPC_D3 compared to NPC_D2. (B) Top 10 enriched GO pathways of upregulated genes in HU-treated NPC_D3 compared to NPC_D2. (C) Top 10 enriched GO pathways of upregulated genes in normally differentiated NPC_D4 compared to NPC_D3. (D) Heatmap showing the expression levels of neural lineage genes and cell division genes in normally differentiated NPCs, HU-arrested NPC_D3 cells, and NPC_D4 following HU release. The color bar represented Z-scores. (E) Top 10 enriched GO pathways of upregulated genes in NPC_D4 with HU release compared to HU-treated NPC_D3. Statistical significance of GO enrichment analysis in (A, B, C, E) is determined using the hypergeometric test and Benjamini-Hochberg false discovery rate (FDR) correction, with a significance threshold of $P < 0.05$.

