## [Peer Review File · The EMBO Journal]

Post-replicative initial expression of PAX6 during neuroectoderm differentiation

Song Hu, Rongao Kou, Zhuojie Su, Guanchen Li, Shutao Qi, Yanxiao Zhang, Haifeng Wang, Ling-ling Chen, and Hongtao Yu

Corresponding author(s): Hongtao Yu (yuhongtao@westlake.edu.cn)

Review Timeline:

Submission Date:	25th Apr 25
Editorial Decision:	5th Jun 25
Revision Received:	16th Aug 25
Editorial Decision:	21st Sep 25
Revision Received:	28th Sep 25
Accepted:	7th Oct 25

Editor: Hartmut Vodermaier

Transaction Report:

Prof. Hongtao Yu
Westlake University
School of Life Sciences
600 Dunyu Road
Hangzhou, Zhejiang
China

5th Jun 2025

Re: EMBOJ-2025-121154
Post-replicative initial expression of PAX6 during neuroectoderm differentiation

Dear Hongtao,

Thank you again for submitting your manuscript on regulated PAX6 expression during neuroectoderm differentiation, and apologies for the delay in getting back to you with a response. We have now received the below-copied feedback from three expert referees, who are somewhat divided in their opinion: while referee 2 is favor of publication with only minor changes, referee 1 is currently not supportive; and referee 3, although in principle appreciating the work, feels that several key issues would need to still be addressed prior to publication.

Faced with these equivocal assessments and upon further referee feedback on each other's reports, I concluded that this work may still become suitable for EMBO Journal publication, but only if the evidence backing key conclusion of the study could be significantly strengthened, along the lines of the referees' suggestions. In particular, referee 3 suggests several avenues for further supporting PAX6 expression only starting in G2 and not already during S phase, and for not being affected by broader S-phase deregulation upon HU treatment. Furthermore, referee 1 proposes analyses of PAX6 target gene expression to clarify if its functional role is also really confined to G2 phase. On the other hand, I would not consider it essential to extend the study to in vivo experiments, but the analysis of publicly available data sets suggested by referee 1 would clearly be helpful to strengthen the study and its conclusions.

Should you be able to satisfactorily address these main concerns, as well as the various more specific points raised in the three reports, we would be open to pursuing a revised manuscript further. Since it is our policy to allow only a single round of major revision, I would very much encourage you to contact me with a revision plan and preliminary point-by-point response already during the early stages of your revision work, so that we could discuss if and how the main points could be resolved, or whether a less completely revised manuscript might alternatively become suitable for publication in one of our sister journals like EMBO Reports or Life Science Alliance. Of course, we would also be open to extension of the default three-months revision period if needed; our 'scooping protection' (meaning that competing work appearing elsewhere in the meantime will not affect our considerations of your study) would of course remain valid also throughout such an extension.

Detailed information on preparing, formatting and uploading a revised manuscript can be found below and in our Guide to Authors. Thank you again for the opportunity to consider this work for The EMBO Journal, and I look forward to hearing from you in due time.

With kind regards,

Hartmut

- 2) Each figure legend must specify
 - size of the scale bars that are mandatory for all micrograph panels
 - the statistical test used to generate error bars and P-values
 - the type error bars (e.g., S.E.M., S.D.)
 - the number (n) and nature (biological or technical replicate) of independent experiments underlying each data point
 - Figures may not include error bars for experiments with $n < 3$; scatter plots showing individual data points should be used instead.
- 3) Revised manuscript text (including main tables, and figure legends for main and EV figures) has to be submitted as editable text file (e.g., .docx format). We encourage highlighting of changes (e.g., via text color) for the referees' reference.
- 4) Each main and each Expanded View (EV) figure should be uploaded as individual production-quality files (preferably in .eps, .tif, .jpg formats). For suggestions on figure preparation/layout, please refer to our Figure Preparation Guidelines: <http://bit.ly/EMBOPressFigurePreparationGuideline>
- 5) Point-by-point response letters should include the original referee comments in full together with your detailed responses to them (and to specific editor requests if applicable), and also be uploaded as editable (e.g., .docx) text files.
- 6) Please complete our Author Checklist, and make sure that information entered into the checklist is also reflected in the manuscript; the checklist will be available to readers as part of the Review Process File. A download link is found at the top of our Guide to Authors: embopress.org/page/journal/14602075/authorguide
- 7) All authors listed as (co-)corresponding need to deposit, in their respective author profiles in our submission system, a unique ORCID identifier linked to their name. Please see our Guide to Authors for detailed instructions.
- 8) Please note that supplementary information at EMBO Press has been superseded by the 'Expanded View' for inclusion of additional figures, tables, movies or datasets; with up to five EV Figures being typeset and directly accessible in the HTML version of the article. For details and guidance, please refer to: embopress.org/page/journal/14602075/authorguide#expandedview
- 9) To facilitate reproducibility and cross-laboratory adoption of methodologies, please structure the Materials & Methods section as outlined in our guide to authors, including a completed Reagents and Tools Table that can be downloaded from our author guidelines as well (<https://www.embopress.org/page/journal/14602075/authorguide#structuredmethods>).
- 10) Digital image enhancement is acceptable practice, as long as it accurately represents the original data and conforms to community standards. If a figure has been subjected to significant electronic manipulation, this must be clearly noted in the figure legend and/or the 'Materials and Methods' section. The editors reserve the right to request original versions of figures and the original images that were used to assemble the figure. Finally, we generally encourage uploading of numerical as well as gel/blot image source data; for details see: embopress.org/page/journal/14602075/authorguide#sourcedata

Revision to The EMBO Journal should be submitted online within 90 days, unless an extension has been requested and approved by the editor; please click on the link below to submit the revision online before 3rd Sep 2025:

Link Not Available

If you choose to alternatively have this study further considered by another EMBO Press publication, please use the following hyperlink to directly transfer the manuscript, optionally with inclusion of referee reports and identities:

Link Not Available

Referee #1:

In this manuscript Hu et al. investigate the coordination between cell cycle progression and cell fate specification of neural progenitor cells. Particularly focusing on the transcription factor Pax6 in an in vitro model of differentiation, they characterize i) the temporal dynamics of neural specification, ii) timing of Pax6 expression during the cell cycle, iii) promoter region regulating

such expression, and iv) molecular pathways connected to such regulatory network. As major claims of this work, they present evidence suggesting that Pax6 is expressed during the G2 phase of the cell cycle and that G2 progression is essential for cell fate transition. Experiments leading to these conclusions include, among others, temporal assessment of protein and transcript expression by immunofluorescence, flow cytometry and transcriptome sequencing.

While characterization of this important transcription factor includes novel and interesting aspects (e.g. promoter regions), I am generally puzzled by the extent of novelty and general conclusions of this work. On the one hand, I do agree that finding expression of Pax6 during G2 is quite surprising and novel, challenging the general assumption that cell fate specification occurs in G1. This is interesting. However, on the other hand, the authors provide no evidence for Pax6 itself to have a functional role during G2. For example, several neuronal markers are known to start to be expressed during G2-M of a progenitor mother cell, but only have a role in their daughters, the neurons. Similarly, Pax6 could be expressed in G2 of a ESC mother cell, but act as transcription factor in the G1 of the daughters NSC. In other words, expression in G2 may have no functional implication within the mother cell itself. Assessing the expression of known downstream targets of Pax6, whether also appearing in G2 (but not before) within the same cell or rather in the subsequent G1 of daughter cells is important in this context.

Also, I am quite puzzled by the limitation of this study to only focus on their in vitro cell line model. I can't tell how much this is robust and generally accepted, but a gazillion papers are available that have performed all sorts of sequencings bulk, single-cell, spatial of neural stem cells in vivo, across developmental times, different species, and so forth. Extending/corroborating their analyses (e.g. as done in extended data 4) in vivo would be quite simple by just using publicly available data. Even better, in situ hybridization of mouse brain sections could confirm expression in G2, upon labelling G2 cells with smart timing of BrdU/EdU, or using a Fucci mouse. In other words, corroboration of G2 expression in vivo is important and seems quite easy but lacking.

Last, but not least, I really do not understand the last experiment and second major claim of this work that: "G2 phase is essential for NPC cell fate transition". Of course it is. I consider it definitionally obvious that blocking the S phase of a "progenitor" prevents the generation of its "progeny". Whether or not mesenchymal signatures emerge as a result, in this particular cell line and in vitro model, has unclear significance.

Referee #2:

In their study, Hu et al. investigate the temporal coupling of cell cycle progression and neural lineage specification during the early differentiation of human embryonic stem cells (ESCs) into neural progenitor cells (NPCs). Using a combination of live-cell imaging (Fucci), flow cytometry, mass spectrometry, and single-cell RNA sequencing, they identify a narrow developmental window during which the key neuroectodermal transcription factor PAX6 is induced, and demonstrate that its initial expression occurs specifically during the G2 phase of the cell cycle. Based on ATAC-seq and lentiviral reporter assays, the authors further characterize the regulatory logic underlying this expression pattern and identify a novel 500-bp upstream region that is both necessary and sufficient to drive G2-specific PAX6 transcription. Using CRISPR-mediated promoter deletions, they show that this regulation is independent of canonical cell cycle regulators such as the MMB-FOXO1 complex, and identify a GC-rich core promoter element that is essential for PAX6 expression during the NPC fate transition, but appears to be regulated by an as-yet unknown mechanism. Finally, by arresting cells in S-phase using hydroxyurea, the authors demonstrate that progression into G2 is required for PAX6 induction and proper NPC fate commitment.

This is a well-executed and conceptually innovative study that rigorously addresses how cell cycle progression intersects with early neural lineage commitment. The finding that neural fate specification occurs in G2 and depends on a defined promoter element upstream of PAX6 is likely to be of broad relevance. Overall, I consider the manuscript suitable for publication in The EMBO Journal following minor revisions to clarify a few mechanistic and interpretational points, as detailed below.

Minor points that should be addressed:

1. There are unwanted leading spaces at the beginning of several sentences throughout the manuscript that should be removed.
2. Although the authors discuss links between cell cycle phase and neurogenesis, they omit key references, e.g. from the work of Federico Calegari's group, who have extensively studied G1-phase lengthening and its role in neurogenic commitment.
3. On page 3, lines 12-13, the sentence "...that can themselves under active cell divisions" appears to lack a verb. It likely should read "that can themselves undergo active cell divisions." Please revise for clarity and grammatical correctness.
4. In Figure 5b, multiple promoter segment deletions (e.g., $\Delta 1$ and $\Delta 2$) appear to reduce reporter activity, with $\Delta 2$ showing the most pronounced effect. On page 3, the authors state that "fluorescence signals were, however, significantly reduced in cells carrying the P500_ $\Delta 2$ promoter." However, it is unclear whether this assessment is based on formal statistical analysis (which may not be feasible with the low sample number) or on effect size. While prioritizing $\Delta 2$ is reasonable given the apparent magnitude of the effect in Figure 5c and the high number of predicted transcription factors in this region, the rationale for focusing on $\Delta 2$ over other deletions should be explained more explicitly, and the use of the term "significant" should be justified or rephrased.
5. Several photomicrographs, including the RFP channel in Figure 3e, appear underexposed and would benefit from increased brightness or contrast to improve interpretability. It would also be helpful to confirm whether image acquisition parameters were consistent across conditions.
6. The schematic model in Figure 7 is helpful but could more clearly highlight the transition from PAX6-/G2 to PAX6+/G2 states and clarify its implications for symmetric fate inheritance after mitosis.

Referee #3:

This manuscript explored a novel and important topic on how cell cycle progression regulates lineage specific gene expression. They used dual-SMAD inhibition to induce PAX6+ neural progenitors from human ESCs and discovered that the initiation of PAX6 expression is after entry of S phase and requires the progression through G2 phase. They also mapped a cis-element in PAX6 promoter that is specifically responsible for cell cycle-dependent regulation.

1. The authors did not rule out the possibility that the activation of PAX6 transcription is in the S phase, which is critical for the mechanistic model and their following search for the transcription factor. The authors mainly used FUCCI reporter to define cell cycle phases and categorized "GEMININ+ and CDT1- with intact nuclei" as G2. However, FUCCI reporter alone cannot clearly separate S and G2 phases. "GEMININ+ and CDT1- with intact nuclei" could also be middle and late S phase cells, especially in ESCs and NPCs that have long S phase. Combine with DNA content, or phase specific markers would be more helpful. Although the authors nicely showed the presence of PAX6 proteins at G2 phase, the key is to find out the initiation time point of upstream transcription event.

1) Many cell cycle phase specific genes' transcription starts at the previous phase, and their proteins accumulate and peak at the current phase.

(<https://www.cell.com/trends/biochemical-sciences/fulltext/S0968-0004%2822%2900148-7>)

PAX6 as a transcription factor could be activated at S phase and accumulate proteins at G2 to regulate its downstream targets transcription at G2 phase. The authors could leverage their sequencing data with known cell-cycle specific transcription events as reference to map PAX6 activation.

2) Figure 2g shows PAX6 mRNA level in 24-52h cells. There is more PAX6 mRNA in S phase than G1. Extended Data Fig.4b, an alternative explanation could be if S phase is the initiation time point of PAX6 transcription, the mRNAs start being transcribed and accumulate towards G2 phase. Thus, there is no obvious accumulation from G1 to S phase, but obvious increase from S to G2 phase. Could the authors show the earliest time point of PAX6 mRNA detection and corresponding cell cycle phase in their single cell data?

3) One alternative experiment is to isolate S and G2 cells by FACS and check the PAX6 promoter accessibility by ATAC-seq.

2. Although the authors mapped a cis-regulatory element that is NPC and cell cycle specific, there is still a lack of understanding on the regulation at the molecular level.

1) The authors checked the top five predicted TFs and focused on EGR1 due to its similar expression pattern to PAX6. The authors should search for TFs expressed in their NPCs from sequencing data. Transcription factor usually is expressed before the activation of their targets.

2) Have the authors checked the reported PAX6 transcription factors SEF, Sp1, and Six3?

(<https://pubmed.ncbi.nlm.nih.gov/11574690/>; <https://pubmed.ncbi.nlm.nih.gov/17066077/>)

3) Figure 3c and d showed that shorter promoter leads to more cells expressing the reporter. Could this be a difference caused by lentivirus titer/transducing efficiency - the longer the promoter, the bigger the plasmid and lower titer?

3. Figure 6.

1) Hydroxyurea induces late G1/early S phase arrest by inhibiting dNTP generation and DNA synthesis. This would affect most S phase events. Could the authors provide evidence that this inhibition does not affect some of the known genes that are transcribed in S phase?

2) Figure 6a shows HU 48h and 24h treatment, could the authors specify which condition is NPC_D3_HU in panel b-d?

3) "These findings indicate that S-phase arrest disrupts normal NPC fate transition, either by preventing progression along the same trajectory or by altering differentiation trajectories."

The authors checked the gene expression of cells arrested at S phase, which could be aberrant gene expression caused by DNA replication stress. Have the authors checked the gene expression after releasing the cells? If S phase arrest leads to prevention of progression on the same trajectory, then release the cells from the arrest, they should observe a portion of cells restore the trajectory. If S phase arrest activates nonneuronal fate gene expression, the authors should show example genes that are absence in NPC_D3 cells, but presence in NPC_D3_HU cells in panel h. Instead, WNT8B, COL2A and COL8A2 are all expressed in NPC samples.

Point-by-point response

Reviewer #1

In this manuscript Hu et al. investigate the coordination between cell cycle progression and cell fate specification of neural progenitor cells. Particularly focusing on the transcription factor Pax6 in an in vitro model of differentiation, they characterize i) the temporal dynamics of neural specification, ii) timing of Pax6 expression during the cell cycle, iii) promoter region regulating such expression, and iv) molecular pathways connected to such regulatory network. As major claims of this work, they present evidence suggesting that Pax6 is expressed during the G2 phase of the cell cycle and that G2 progression is essential for cell fate transition. Experiments leading to these conclusions include, among others, temporal assessment of protein and transcript expression by immunofluorescence, flow cytometry and transcriptome sequencing.

While characterization of this important transcription factor includes novel and interesting aspects (e.g. promoter regions), I am generally puzzled by the extent of novelty and general conclusions of this work. On the one hand, I do agree that finding expression of Pax6 during G2 is quite surprising and novel, challenging the general assumption that cell fate specification occurs in G1. This is interesting. However, on the other hand, the authors provide no evidence for Pax6 itself to have a functional role during G2. For example, several neuronal markers are known to start to be expressed during G2-M of a progenitor mother cell, but only have a role in their daughters, the neurons. Similarly, Pax6 could be expressed in G2 of a ESC mother cell, but act as transcription factor in the G1 of the daughters NSC. In other words, expression in G2 may have no functional implication within the mother cell itself. Assessing the expression of known downstream targets of Pax6, whether also appearing in G2 (but not before) within the same cell or rather in the subsequent G1 of daughter cells is important in this context.

Response: We thank the reviewer for this great suggestion. Previous studies have identified PAX6 target genes involved in neurogenesis across various model organisms using ChIP-seq experiments (PMID: 27462442; PMID: 21617155; PMID: 23342162; PMID: 26138486). However, these studies were conducted on tissues that had already undergone neural induction, in which NPCs were in a relatively mature state. To date, there have been no reports characterizing PAX6 target genes during the early stages of human neural induction. To address this, we constructed PAX6 KO ESCs and differentiated them into NPCs. We then conducted RNA-seq and CUT&Tag experiments and identified 45 PAX6 target genes during early NPC differentiation. Our scRNA-seq-based cell cycle analysis revealed that several key PAX6 target genes, including *PAX6*, *MECOM*, and *CDH6* were transcriptionally activated in the G2 phase following the onset of *PAX6* expression. Notably, the autoregulatory mechanism of *PAX6* has been reported previously. Our results have now demonstrated that this self-regulation is initiated in G2 upon the onset of *PAX6* expression. These findings have greatly strengthened our conclusion that PAX6 begins to exert functions during the G2 phase of the mother cell. We have included these new results in Fig. EV5 of the revised manuscript.

Also, I am quite puzzled by the limitation of this study to only focus on their in vitro cell line model. I can't tell how much this is robust and generally accepted, but a gazillion papers are available that have performed all sorts of sequencings bulk, single-cell, spatial of neural stem cells in vivo, across developmental times, different species, and so forth.

Extending/corroborating their analyses (e.g. as done in extended data 4) in vivo would be quite simple by just using publicly available data. Even better, in situ hybridization of mouse brain sections could confirm expression in G2, upon labelling G2 cells with smart timing of BrdU/EdU, or using a FUCCI mouse. In other words, corroboration of G2 expression in vivo is important and seems quite easy but lacking.

Response: We again thank the reviewer for this great suggestion. The human neural plate first appears at 18 days after fertilization (PMID: 15939212). We thus analyzed the publicly available single-cell RNA sequencing datasets of early human embryos: PCW3 (PMID: 37192616) or CS12 (PMID: 33723434). Our analysis revealed that *PAX6*⁺ cells were indeed enriched in G2/M across neuroectoderm lineages in human embryos. This is consistent with our *in vitro* findings. These new data have been included in Figs. 2H,I and EV4E,F in the revised manuscript.

Last, but not least, I really do not understand the last experiment and second major claim of this work that: "G2 phase is essential for NPC cell fate transition". Of course it is. I consider it definitionally obvious that blocking the S phase of a "progenitor" prevents the generation of its "progeny". Whether or not mesenchymal signatures emerge as a result, in this particular cell line and in vitro model, has unclear significance.

Response: We agree with the reviewer that our initial statement was confusing. We have revised our statement as "Blocking the cell cycle at S phase prevents *PAX6* expression." We have emphasized that the emergence of mesenchymal signatures after HU treatment only indicates impaired cell fate decisions after cell cycle arrest, which does not normally occur under physiological conditions *in vivo*.

Reviewer #2

*In their study, Hu et al. investigate the temporal coupling of cell cycle progression and neural lineage specification during the early differentiation of human embryonic stem cells (ESCs) into neural progenitor cells (NPCs). Using a combination of live-cell imaging (FUCCI), flow cytometry, mass spectrometry, and single-cell RNA sequencing, they identify a narrow developmental window during which the key neuroectodermal transcription factor *PAX6* is induced, and demonstrate that its initial expression occurs specifically during the G2 phase of the cell cycle. Based on ATAC-seq and lentiviral reporter assays, the authors further characterize the regulatory logic underlying this expression pattern and identify a novel 500-bp upstream region that is both necessary and sufficient to drive G2-specific *PAX6* transcription. Using CRISPR-mediated promoter deletions, they show that this regulation is independent of canonical cell cycle regulators such as the MMB-FOXO1 complex, and identify a GC-rich core promoter element that is essential for *PAX6* expression during the NPC fate transition, but appears to be regulated by an as-yet unknown mechanism. Finally, by arresting cells in S-phase using hydroxyurea, the authors demonstrate that progression into G2 is required for *PAX6* induction and proper NPC fate commitment.*

*This is a well-executed and conceptually innovative study that rigorously addresses how cell cycle progression intersects with early neural lineage commitment. The finding that neural fate specification occurs in G2 and depends on a defined promoter element upstream of *PAX6* is likely to be of broad relevance. Overall, I consider the manuscript suitable for publication in *The**

EMBO Journal following minor revisions to clarify a few mechanistic and interpretational points, as detailed below.

Response: We thank the reviewer for his/her positive comments.

Minor points that should be addressed:

1. There are unwanted leading spaces at the beginning of several sentences throughout the manuscript that should be removed.

Response: These formatting errors have been corrected in our revised manuscript.

2. Although the authors discuss links between cell cycle phase and neurogenesis, they omit key references, e.g. from the work of Federico Calegari's group, who have extensively studied G1-phase lengthening and its role in neurogenic commitment.

Response: We apologize for this oversight. These references, including the work from the Calegari group, have been cited in the revised manuscript.

3. On page 3, lines 12-13, the sentence "...that can themselves under active cell divisions" appears to lack a verb. It likely should read "that can themselves undergo active cell divisions." Please revise for clarity and grammatical correctness.

Response: This error will be corrected in our revised manuscript.

4. In Figure 5b, multiple promoter segment deletions (e.g., $\Delta 1$ and $\Delta 2$) appear to reduce reporter activity, with $\Delta 2$ showing the most pronounced effect. On page 3, the authors state that "fluorescence signals were, however, significantly reduced in cells carrying the P500_ $\Delta 2$ promoter." However, it is unclear whether this assessment is based on formal statistical analysis (which may not be feasible with the low sample number) or on effect size. While prioritizing $\Delta 2$ is reasonable given the apparent magnitude of the effect in Figure 5c and the high number of predicted transcription factors in this region, the rationale for focusing on $\Delta 2$ over other deletions should be explained more explicitly, and the use of the term "significant" should be justified or rephrased.

Response: Thanks for the suggestion. The assessment of reporter activities is based on effect size. We have explained our rationale more explicitly and changed the term from "significantly" to "greatly" in our revised manuscript.

5. Several photomicrographs, including the RFP channel in Figure 3e, appear underexposed and would benefit from increased brightness or contrast to improve interpretability. It would also be helpful to confirm whether image acquisition parameters were consistent across conditions.

Response: Thanks for the suggestion. We have adjusted the brightness and contrast to improve the quality of these images in the revised manuscript. The images were acquired with the same parameters across different conditions.

6. The schematic model in Figure 7 is helpful but could more clearly highlight the transition from PAX6-/G2 to PAX6+/G2 states and clarify its implications for symmetric fate inheritance after mitosis.

Response: Thanks for the suggestion. We have modified our model (Fig. 7) to highlight the cell fate transition in the revised manuscript.

Reviewer #3

This manuscript explored a novel and important topic on how cell cycle progression regulates lineage specific gene expression. They used dual-SMAD inhibition to induce PAX6+ neural progenitors from human ESCs and discovered that the initiation of PAX6 expression is after entry of S phase and requires the progression through G2 phase. They also mapped a cis-element in PAX6 promoter that is specifically responsible for cell cycle-dependent regulation.

1. The authors did not rule out the possibility that the activation of PAX6 transcription is in the S phase, which is critical for the mechanistic model and their following search for the transcription factor. The authors mainly used FUCCI reporter to define cell cycle phases and categorized "GEMININ+ and CDT1- with intact nuclei" as G2. However, FUCCI reporter alone cannot clearly separate S and G2 phases. "GEMININ+ and CDT1- with intact nuclei" could also be middle and late S phase cells, especially in ESCs and NPCs that have long S phase. Combine with DNA content, or phase specific markers would be more helpful. Although the authors nicely showed the presence of PAX6 proteins at G2 phase, the key is to find out the initiation time point of upstream transcription event.

Response: Our FACS data have clearly demonstrated that the PAX6 protein emerges in G2 cells. Because the function of PAX6 depends on the protein, the expression of the PAX6 protein should be more direct than the PAX6 transcripts in determining the neuroectoderm cell fate. With this said, our single-cell RNA sequencing data indicated that the PAX6⁺ cells were enriched in G2 (Fig. 2G), consistent with our reporter assays. On the other hand, we did observe a minor fraction of PAX6⁺ cells in S phase. In addition, the S-G2 transition is not clearly marked by definitive molecular events. It is thus difficult to define the exact S-G2 boundary. We have therefore revised our conclusions to indicated that PAX6 transcription is enhanced in G2 in the revised manuscript.

1) Many cell cycle phase specific genes' transcription starts at the previous phase, and their proteins accumulate and peak at the current phase. (<https://www.cell.com/trends/biochemical-sciences/fulltext/S0968-0004%2822%2900148-7>). PAX6 as a transcription factor could be activated at S phase and accumulate proteins at G2 to regulate its downstream targets transcription at G2 phase. The authors could leverage their sequencing data with known cell-cycle specific transcription events as reference to map PAX6 activation.

Response: Thanks for the suggestion. We analyzed our scRNA-seq data and used **CellCycleScoring** (PMID: 26430063) to discriminate between different cell cycle phases. Based on our data, most of the PAX6⁺ cells were at the G2 phase, with a minor population of PAX6⁺ cells at S phase. Thus, PAX6 transcription is clearly elevated in the G2 phase.

2) Figure 2g shows *PAX6* mRNA level in 24-52h cells. There is more *PAX6* mRNA in S phase than G1. Extended Data Fig.4b, an alternative explanation could be if S phase is the initiation time point of *PAX6* transcription, the mRNAs start being transcribed and accumulate towards G2 phase. Thus, there is no obvious accumulation from G1 to S phase, but obvious increase from S to G2 phase. Could the authors show the earliest time point of *PAX6* mRNA detection and corresponding cell cycle phase in their single cell data?

Response: We agree with this excellent point. We have examined the cell cycle distribution of *PAX6* transcripts at NPC_48h, representing the earliest stage of *PAX6* transcriptional activation. Our analysis indicated a marked enrichment of *PAX6* expression in G2-phase cells. This result has been included as Fig. EV5F in the revised manuscript.

3) One alternative experiment is to isolate S and G2 cells by FACS and check the *PAX6* promoter accessibility by ATAC-seq.

Response: Thanks for the suggestion. We have performed scATAC-seq experiments. Integration of scRNA-seq and scATAC-seq analysis indicated that there was no obvious difference in the chromatin accessibility of the *PAX6* P500 promoter across G1, S, and G2/M phases, regardless of whether cells were *PAX6*-positive or negative (Fig. EV7C). These findings suggest that the enhanced *PAX6* transcription in G2 does not appear to be driven by increased chromatin accessibility.

2. Although the authors mapped a cis-regulatory element that is NPC and cell cycle specific, there is still a lack of understanding on the regulation at the molecular level.

1) The authors checked the top five predicted TFs and focused on *EGR1* due to its similar expression pattern to *PAX6*. The authors should search for TFs expressed in their NPCs from sequencing data. Transcription factor usually is expressed before the activation of their targets.

Response: Thanks for the suggestion. We have searched for potential upstream TFs of *PAX6* that were expressed before *PAX6* expression based on our sequencing data. However, genes that were upregulated prior to *PAX6* expression—specifically, those showing increased expression at NPC_D2 compared to NPC_D1 ($\log_2\text{FC} > 1.5$, $p < 0.05$)—showed no overlap with the top 50 TF candidates predicted by the JASPAR database to bind the *PAX6* P500 promoter region. This lack of intersection and negative results of other tested TFs in our study suggested that *PAX6* expression might be regulated by a novel transcription factor or by the redundant actions of the tested transcription factors.

2) Have the authors checked the reported *PAX6* transcription factors *SEF*, *Sp1*, and *Six3*? (<https://pubmed.ncbi.nlm.nih.gov/11574690/>; <https://pubmed.ncbi.nlm.nih.gov/17066077/>)

Response: Thanks for the suggestion. Based on our bulk RNA-seq data, *SEF* and *SP1* were already expressed in ESCs, while *SIX3* was absent from ESCs but expressed from Day 1 of neural induction (Fig. EV8C,D). Moreover, *SIX3* was among the top hits in a CRISPR screen for *PAX6* regulators (PMID: 36381608), while *SEF* and *SP1* were not recovered as screen hits. Based on these findings, we suspected that *SIX3* might be an upstream TF that drives the

expression of *PAX6*. Unfortunately, as shown in Fig. EV8F,G, *SIX3* KO did not affect the expression of *PAX6*.

3) *Figure 3c and d showed that shorter promoter leads to more cells expressing the reporter. Could this be a difference caused by lentivirus titer/transducing efficiency - the longer the promoter, the bigger the plasmid and lower titer?*

Response: We agree that different promoter sizes could affect the lentivirus titer/transducing efficiency, with the larger plasmids potentially having lower titers. We have mentioned this caveat in the revised manuscript. Our conclusion that the P500 promoter is the minimal promoter is not affected by this caveat. We have, however, removed our discussion about the putative inhibitory element in this region in the revised manuscript.

3. Figure 6.

1) *Hydroxyurea induces late G1/early S phase arrest by inhibiting dNTP generation and DNA synthesis. This would affect most S phase events. Could the authors provide evidence that this inhibition does not affect some of the known genes that are transcribed in S phase?*

Response: Thanks for the suggestion. We analyzed our bulk RNA-seq data and checked the expression of known S-phase transcribed genes (PMID: 15362224), such as *CCNE1*, *CDC25A*, *CDK2*, *CDC6*, and *POLA2*. Our results showed that those genes remained enriched in S phase following HU-induced arrest (Fig. 6H), indicating that HU treatment did not interfere with the transcription of canonical S-phase genes.

2) *Figure 6a shows HU 48h and 24h treatment, could the authors specify which condition is NPC_D3_HU in panel b-d?*

Response: We have specified this information in both the figure and the figure legend.

3) *"These findings indicate that S-phase arrest disrupts normal NPC fate transition, either by preventing progression along the same trajectory or by altering differentiation trajectories." The authors checked the gene expression of cells arrested at S phase, which could be aberrant gene expression caused by DNA replication stress. Have the authors checked the gene expression after releasing the cells? If S phase arrest leads to prevention of progression on the same trajectory, then release the cells from the arrest, they should observe a portion of cells restore the trajectory. If S phase arrest activates nonneuronal fate gene expression, the authors should show example genes that are absent in NPC_D3 cells, but presence in NPC_D3_HU cells in panel h. Instead, WNT8B, COL2A and COL8A2 are all expressed in NPC samples.*

Response: Thanks for the suggestion. We did not observe any pathways related to cellular stress responses among the significantly enriched GO terms (adjusted $p < 0.05$) that were upregulated following HU treatment, suggesting that the observed transcriptional changes are unlikely to result from non-specific stress-induced effects, but rather reflect S-phase arrest. To further support this conclusion, we have included the expression profiles of neuroectoderm lineage genes in the revised manuscript (Fig. 6H), highlighting those that were highly expressed in the NPC_D3 group but showed reduced expression in the NPC_D3_HU_24h group.

As suggested, we performed the HU release experiment to assess the reversibility of the aberrant cell fate induced by S-phase arrest. We subjected the NPC_D3_HU_24h samples to a 24-hour HU release, generating NPC_D4_HU_release samples. Our bulk RNA-seq data showed that genes upregulated in NPC_D4_HU_release compared to NPC_D3_HU_24h were enriched in brain development-related pathways, including *PAX6*, *SP8*, *FOXG1*, *CDH6*, and *EMX2*, and pathways associated with mitotic nuclear division, such as *MKI67*, *CENPF*, *PLK1*, *CCNB1*, and *AURKA* (Fig. EV10D,E). These findings suggest that, following the release from HU-induced S-phase arrest, NPCs could resume cell cycle progression and restore the neural differentiation trajectory. We note, however, that the expression levels of neurodevelopmental genes in NPC_D4_HU_release samples did not reach those observed in NPC_D4_Ctrl, indicating that the deviation from the neurogenic trajectory caused by HU treatment could not be fully reversed following the removal of HU. Taken together, these results suggest a requirement for G2 phase progression in NPC fate transition. S-phase arrest not only halts neural differentiation but also redirects cells toward mesenchymal lineages, underscoring the critical role of cell cycle dynamics in dictating progenitor cell fate.

Prof. Hongtao Yu
Westlake University
School of Life Sciences
600 Dunyu Road
Hangzhou, Zhejiang
China

21st Sep 2025

Re: EMBOJ-2025-121154R
Post-replicative initial expression of PAX6 during neuroectoderm differentiation

Dear Hongtao,

Thank you for submitting your revised manuscript. Two of the original referees have now looked into it once more, and given their overall satisfaction with the revisions (see below), we would be happy to accept your study for publication in The EMBO Journal, pending adequate addressing of the following remaining editorial issues:

- Importantly, our routine pre-acceptance image checks indicate that one of the blots in Figure 5H appears to completely lack contrast/background features, and looks non-natural even in the provided source data images. This needs to be satisfactorily clarified, ideally by providing additional, unprocessed raw data also showing larger areas rather than just lane crops.
- In the Data Availability section, please make sure to provide simple, final URLs for the deposited datasets, and also remove reviewer tokens at this stage, instead ensuring public accessibility of the datasets at this point.
- Please carefully go through the reference list and make sure that each reference is complete with citation year, volume, and page/locator numbers (currently missing in several of them), and that alternative DOI information is only included for pre-publication manuscripts that do not have a formal citation description yet.
- Finally, please provide suggestions for a short 'blurb' text prefacing and summing up the conceptual aspect of the study in two sentences (max. 250 characters), followed by 3-5 one-sentence 'bullet points' with brief factual statements of key results of the paper; they will form the basis of an editor-written 'Synopsis' accompanying the online version of the article. Please also upload a synopsis image, which can be used as a "visual title" for the synopsis section of your paper. The image (maybe based on a condensed/simplified version of Figure 7?) should be in PNG or JPG format, and please make sure that it remains in the modest dimensions of (exactly) 550 pixels wide and 300-600 pixels high.

I am therefore returning the manuscript to you for an additional round of minor revision, hoping that you will be able to clarify the remaining issues and resubmit all materials in a satisfactory manner. Please do not hesitate to contact me with any questions that you may have in this regard.

With kind regards,

Hartmut

*** PLEASE NOTE: All revised manuscripts are subject to initial checks for completeness and adherence to our formatting guidelines. Revisions may be returned to the authors and delayed in their editorial re-evaluation if they fail to comply with the following requirements (see also our Guide to Authors for further information):

- 1) Every manuscript requires a Data Availability section (even if only stating that no deposited datasets are included). Primary datasets or computer code produced in the current study have to be deposited in appropriate public repositories prior to resubmission, and reviewer access details provided in case that public access is not yet allowed. Further information: embopress.org/page/journal/14602075/authorguide#dataavailability
- 2) Each figure legend must specify
 - size of the scale bars that are mandatory for all micrograph panels

- the statistical test used to generate error bars and P-values
- the type error bars (e.g., S.E.M., S.D.)
- the number (n) and nature (biological or technical replicate) of independent experiments underlying each data point
- Figures may not include error bars for experiments with $n < 3$; scatter plots showing individual data points should be used instead.

9) To facilitate reproducibility and cross-laboratory adoption of methodologies, please structure the Materials & Methods section as outlined in our guide to authors, including a completed Reagents and Tools Table that can be downloaded from our author guidelines as well (<https://www.embopress.org/page/journal/14602075/authorguide#structuredmethods>).

10) Digital image enhancement is acceptable practice, as long as it accurately represents the original data and conforms to community standards. If a figure has been subjected to significant electronic manipulation, this must be clearly noted in the figure legend and/or the 'Materials and Methods' section. The editors reserve the right to request original versions of figures and the original images that were used to assemble the figure. Finally, we generally encourage uploading of numerical as well as gel/blot image source data; for details see: embopress.org/page/journal/14602075/authorguide#sourcedata

In the interest of ensuring the conceptual advance provided by the work, we recommend submitting a revision within 3 months (20th Dec 2025). Please discuss the revision progress ahead of this time with the editor if you require more time to complete the revisions. Use the link below to submit your revision:

Link Not Available

Referee #2:

The authors have addressed all my points.

Referee #3:

The authors put a great deal of extra work and important improvements to the manuscript by conducting additional analysis on sequencing data, comparison to human fetal tissue expression profile, testing additional transcription factors using KO iPSC

lines. Although it's disappointing that the TF responsible for PAX6 expression at G2 phase is still unknown, we appreciate the authors efforts. Overall, they have addressed all previous concerns.

Referee #3:

The authors put a great deal of extra work and important improvements to the manuscript by conducting additional analysis on sequencing data, comparison to human fetal tissue expression profile, testing additional transcription factors using KO iPSC lines. Although it's disappointing that the TF responsible for PAX6 expression at G2 phase is still unknown, we appreciate the authors efforts. Overall, they have addressed all previous concerns.

Response: We are grateful for the constructive feedback and are pleased that the reviewer finds our revisions have addressed the previous concerns.